# Sphingosine-1-Phosphate-derived 2-Hexadecenal is a central mediator of ocular neovascularization by inhibiting Sphingosine-1-Phosphate receptor 5

Xin Qian[1,2], Rui Ge[1], Yinteng Chu[2], Tian Kuang[3], Xin Zhang[1], Katrin Bennewitz[1], Bowen Lou[1], Weijie Hao[4], Volker Ast[5], Glynis Klinke[6], Gernot Poschet[6,7], Jakob Morgenstern[8], Thomas Fleming[8], Ingrid Hausser[9], Julia Szendroedi[8], Peter Paul Nawroth[10] & Jens Kroll[1] ✉

Sphingosine-1-phosphate (S1P) is a crucial sphingolipid mediator in vasculature and neovascular eye diseases by controlling angiogenesis, inflammation and fibrosis. Five S1P receptors (S1PRs) are key therapeutic targets, with several S1PR-targeted drugs already in clinical use or trials. However, the vascular function of its major metabolic product, the reactive lipid aldehyde 2-hexadecenal (2-HD), remains unexplored. Here, we show that loss of the aldehyde dehydrogenase ALDH3B1 impairs 2-HD detoxification and leads to retinal vascular abnormalities in zebrafish, without affecting the trunk vasculature. Mechanistically, multi-omics analyses reveal that 2-HD accumulation disrupts iron homeostasis and induces ferroptosis by directly interacting with S1PR5. This finding is supported by integrative analyses of single-cell RNA sequencing and RNA sequencing from human neovascular retinal samples, identifying S1PR5 as a clinically relevant target. These findings uncover a previously unrecognized role of S1P derived 2-HD in vasculature and retinal vascular homeostasis, suggesting that targeting S1PR5 could offer a therapeutic strategy for diabetic retinopathy.

Ocular neovascularization, including retinopathy of prematurity (ROP) in pediatric populations, diabetic retinopathy (DR) in working age populations and age-related macular degeneration (AMD) in the elderly, is the leading cause of visual impairment in patients with eye diseases[1]. Inadequate metabolic supply in these diseases triggers complex processes such as angiogenesis, edema, inflammation, cell death, and fibrosis, leading to disorganized angiogenesis and, in severe cases, fibrotic scarring or retinal detachment[2,3]. Current

[1]Department of Vascular Biology, European Center for Angioscience (ECAS), Medical Faculty Mannheim, Heidelberg University, Mannheim, Germany. [2]Department of Vascular Surgery, Renji Hospital, School of Medicine, Shanghai Jiaotong University, Shanghai, China. [3]Department of Gastrointestinal Surgery, Renji Hospital, School of Medicine, Shanghai Jiaotong University, Shanghai, China. [4]State Key Laboratory of Oncogenes and Related Genes, Renji-Med X Clinical Stem Cell Research Center, Ren Ji Hospital, School of Medicine, Shanghai Jiao Tong University, Shanghai, China. [5]NGS Core Facility, Medical Faculty Mannheim, Heidelberg University, Mannheim, Germany. [6]Metabolomics Core Technology Platform, NGS Core Facility, Medical Faculty MannheimCentre for Organismal Studies, Heidelberg University, Heidelberg, Germany. [7]Cluster of Excellence GreenRobust, Heidelberg University, Heidelberg, Germany. [8]Department of Internal Medicine I and Clinical Chemistry, Heidelberg University Hospital, Heidelberg, Germany. [9]Institute of Pathology IPH, EM Lab, Heidelberg University Hospital, Heidelberg, Germany. [10]Medical Clinic and Polyclinic II, University Hospital Dresden, Dresden, Germany. ✉e-mail: jens.kroll@medma.uni-heidelberg.de

therapeutic strategies, such as laser photocoagulation and vascular endothelial growth factor (VEGF) neutralization, have revolutionized the treatment of DR and AMD, markedly improving visual outcomes. However, despite their success, these approaches still carry limitations, including high costs, the need for repeated intravitreal injections, and potential complications such as retinal detachment and endophthalmitis. This underscores the need for the development of novel therapeutic strategies with fewer side effects and limitations.

Sphingosine-1-phosphate (S1P), a pleiotropic lipid mediator with significant roles in the vascular and immune systems, has emerged as a crucial topic in the research of neovascular ocular diseases[2,4,5]. Previous studies have demonstrated that S1P contributes to both retinal and choroidal neovascularization (CNV) in AMD and DR, and that pharmacological modulation of S1P signaling can effectively attenuate pathological angiogenesis[5]. The biological actions of S1P are mediated by five G protein-coupled receptors (S1PR1–5 receptors) that are highly expressed in the human retina[2]. In contrast to VEGF, S1P facilitates the formation of stable blood vessels and strengthens endothelial barrier integrity[2,6]. S1P signaling also has the potential to modulate immune and inflammatory responses, as evidenced by the use of fingolimod, an S1PR modulator, to reduce ocular inflammation in multiple sclerosis (MS) patients[7]. In addition, S1P signaling has been implicated in fibrotic processes associated with neovascular ocular diseases. Anti-S1P antibodies showed efficacy in reducing collagen deposition in a mouse model of choroidal neovascularization CNV[2,8]. Despite these advances, the pathophysiological contributions of S1P metabolites, particularly 2-hexadecenal (2-HD), a major product of S1P degradation, remained unexplored in the context of neovascular ocular diseases.

2-HD is an unsaturated long-chain fatty aldehyde that can be produced enzymatically or non-enzymatically by the free-radical mediated cleavage of S1P[9,10], and has been reported to induce apoptosis in fibroblasts and cancer cells[11–15]. The detoxification of 2-HD involves oxidation to trans-2-hexadecenoic acid followed by the addition of acetyl CoA; however, the physiological impact of this metabolic pathway is not well characterized[12,16,17]. Our previous studies have indicated that fatty acids aldehydes are metabolized by specific enzymatic systems, including aldehyde dehydrogenases (ALDH)[18,19]. The absence of these detoxifying enzymes led to an accumulation of aldehydes, leading to retinal neovascularization. Well-studied examples such as 4-Hydroxynonenal (4-HNE) and trans, trans-2,4-decadienal (tt-DDE), which are detoxified by Aldh3a1[18] and Aldh9a1b[19], respectively, are known to affect glucose regulation and contribute to DR. Similarly, the known detoxification system for 2-HD involves ALDH3A2 and ALDH3B1[20–22]. They exhibit distinct cellular localizations and functions, with ALDH3A2 mainly targeting C16 aldehydes in the endoplasmic reticulum, and ALDH3B1 dealing with oxidative stress-induced aldehydes at the plasma membrane[20,21]. Intriguingly, more than 130 mutations in the ALDH3A2 enzyme have been linked to Sjögren−Larsson syndrome, a rare autosomal recessive disorder that may result from the accumulation of fatty aldehydes, particularly 2-HD[23,24]. Our previous studies analyzing both aldh3a2 and aldh3b1 in zebrafish models of retinal neovascularization revealed a significant upregulation of aldh3b1[18], suggesting its compensatory role in the ocular neovascularization by facilitating detoxification of 2-HD. This led to the hypothesis that ALDH3B1 is responsible for the detoxification of 2-HD, and that enzyme dysfunction may lead to 2-HD accumulation, altered S1P signaling and contribute to the pathology of ocular neovascular diseases.

Given the implicated relationship between 2-HD and ALDH3B1 and the significant potential of 2-HD in ocular neovascular diseases, we established an aldh3b1 knockout zebrafish line to investigate its role in the ocular vasculature. Our study confirmed 2-HD as a substrate for Aldh3b1 and demonstrated that its accumulation disrupts the S1pr5

receptor, induces ferroptosis and contributes to the pathological progression of ocular neovascularization. Further analysis of human retinal single-cell sequencing and RNA-seq data corroborated our findings in human DR and highlighted S1PR5 modulation, influenced by 2-HD accumulation, as a promising and novel therapeutic target for DR.

## Results

### Generation and validation of aldh3b1⁻/⁻ zebrafish using CRISPR/Cas9 technology

To gain insight into the role of 2-HD in vascular function, aldh3b1 was knocked out in the Tg(fli1:EGFP) zebrafish line using CRISPR technology. The evolutionary conservation of Aldh3b1 was investigated by phylogenetic analysis and alignment of amino acid sequences. The resulting phylogenetic bootstrap tree showed conserved evolutionary relationships among representative vertebrate animal species (Fig. 1A). Moreover, the amino acid sequence of zebrafish Aldh3b1 exhibited high similarity to those of human and mouse ALDH3B1, including the conservation of active sites, suggesting a similar biological function in these organisms (Fig. 1B).

The aldh3b1 CRISPR knockout was designed to target exon2. Sequencing of the genomic DNA from aldh3b1⁻/⁻ zebrafish indicated a 16 bp insertion that was predicted to alter the amino acid sequence (Fig. 1C). The potential function of aldh3b1 was further explored by analyzing its expression during larval development and in adult organs. Expression levels of aldh3b1 mRNA remained unchanged from 1 to 5 days post-fertilization (dpf) (Fig. 1D) and were highest in the brain and eyes (Fig. 1E), indicating a pivotal role in these tissues.

The successful establishment of the aldh3b1⁻/⁻ zebrafish line was validated at the DNA, protein, and functional levels. Genotyping identified the presence of the insertion on a genotyping-PCR gel (Fig. 1F), which was consistent with Mendelian inheritance in the F2 generation (Fig. 1G). Immunoblot analysis confirmed the absence of Aldh3b1 protein in the eyes of aldh3b1⁻/⁻ zebrafish (Fig. 1H). Furthermore, a reduction in total ALDH enzyme activity using the substrate acetaldehyde showed the functional loss of Aldh3b1 (Fig. 1I).

Subsequent analysis revealed that aldh3b1 knockout did not affect the survival of adult zebrafish maintained under standard conditions up to 15 months of age (Fig. 1J), nor did it affect the length and weight of adult zebrafish (Fig. 1K, L). Collectively, these findings suggested the successful generation of aldh3b1 knockout zebrafish and the functional significance of Aldh3b1 in zebrafish.

### Microvascular alterations in aldh3b1⁻/⁻ mutants

Our previous studies have identified a significant upregulation of aldh3b1 in ocular neovascular zebrafish models[18]. Therefore, vascular changes in aldh3b1⁻/⁻ zebrafish were investigated, particularly in relation to neovascular retinopathy. In larvae, a significant increase in the number of hyaloid sprouts and branches was observed in aldh3b1⁻/⁻ 5dpf larvae (Fig. 2A–C), while no significant changes were detected in the trunk vasculature, including hyperbranches and abnormal intersegmental vessels (Supplementary Fig. S1A−C). Vascular alterations in the adult zebrafish retina were similar to the hyaloid vasculature with increased retinal sprouts and branches (Fig. 2D−F).

Given the suggested association between aldehydes and glucose metabolism, it was hypothesized that the microvascular alterations were a result of impaired glucose metabolism due to the absence of aldh3b1. Therefore, glucose levels and glucose metabolism were analyzed in aldh3b1⁻/⁻ zebrafish. No significant change in tissue glucose levels was observed in larvae (Supplementary Fig. S1D), and the loss of aldh3b1 did not affect fasting or 2-hour postprandial blood glucose levels in adult fish (Supplementary Fig. S1E). Further examination of the glucose metabolic pathway revealed unchanged expression of insulin signaling components − including insulin (ins), insulin receptor

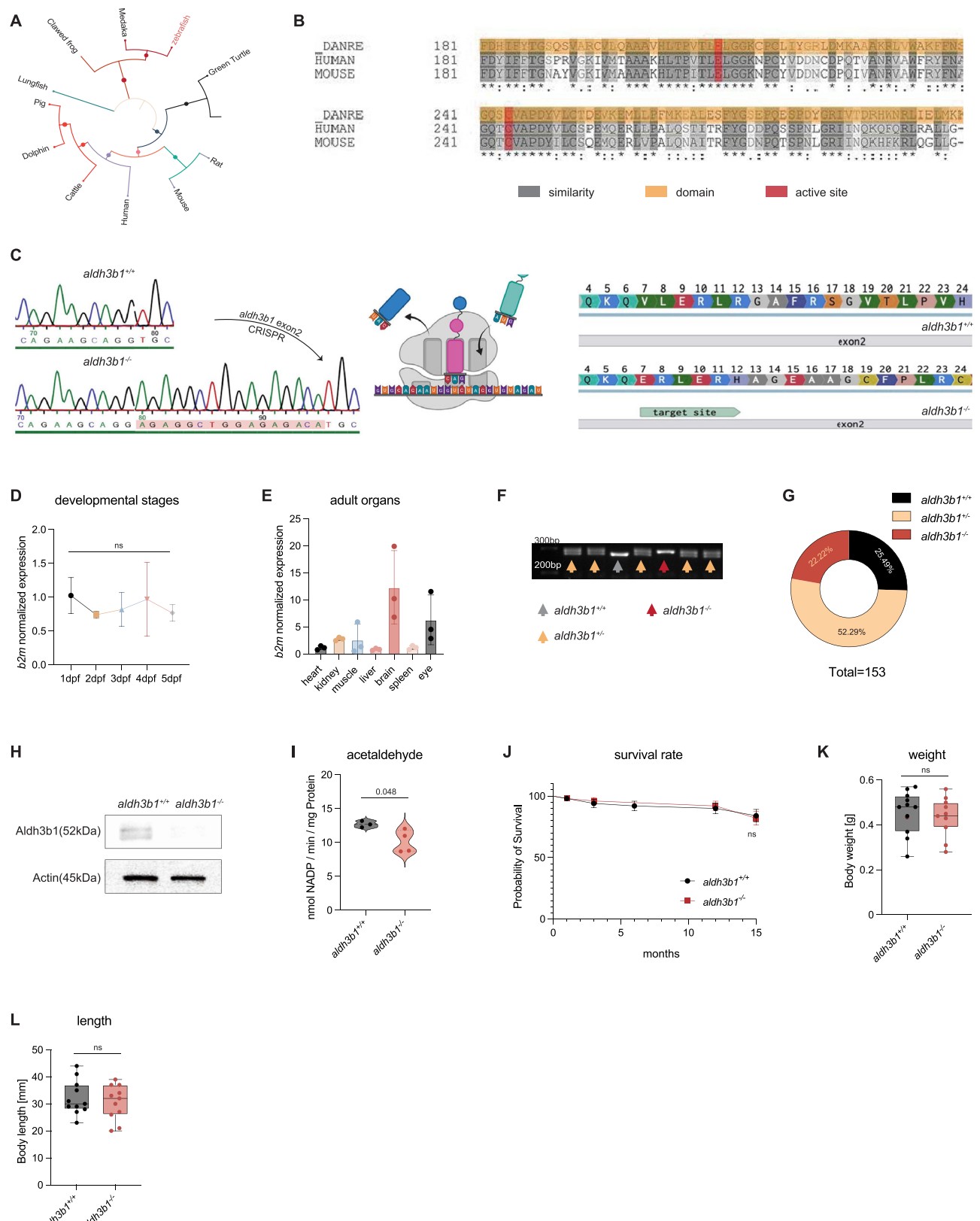

a (*insra*), *insulin receptor* b (*insrb*), and *pancreatic and duodenal homeobox 1* (*pdx1*) in *aldh3b1^-/-* larvae (Supplementary Fig. S1F). In addition, analyses did not reveal any significant changes in glucose metabolism-related signaling pathways, such as glycolysis, gluconeogenesis, and glucose transport, in *aldh3b1^-/-* zebrafish (Supplementary Fig. S1G–I).

In summary, *aldh3b1* knockout induced pathological angiogenesis in the retina that was independent of glucose metabolism dysregulation. This discovery adds a new dimension to our understanding of microvascular complications and elucidated that *aldh3b1* plays an essential role in maintaining retinal vascular function beyond metabolic mechanisms.

**Fig. 1 | Generation and validation of *aldh3b1⁻/⁻* zebrafish using CRISPR/Cas9 technology. A** Phylogenetic bootstrap tree illustrated the evolutionary relationship of *aldh3b1* across representative vertebrate species. **B** Alignment of the Aldh3b1 amino acid sequence revealed high similarity, including conserved active site and domain among the indicated species. **C** Targeted CRISPR/Cas9 mutagenesis of *aldh3b1* in exon 2 resulted in 16 bp insertions, altering the amino acid sequence. Created in BioRender. Bennewitz, K. (2026) https://BioRender.com/3e7287f. **D, E** Quantitative analysis showed that *aldh3b1* mRNA expression levels remain consistent through various developmental stages of zebrafish larvae and was predominantly expressed in the brain and eyes of adult zebrafish (*n* = 3 for each stage and organ). **F, G** *aldh3b1* genotyping can be clearly distinguished on agarose gels (**F**) and was in line with the mendelian inheritance in the first generation of F2

(**G**). **H** Immunoblot analysis confirmed the absence of Aldh3b1 protein in *aldh3b1⁻/⁻* eye. **I** Total ALDH enzyme activity was reduced in *aldh3b1⁻/⁻* larvae using the substrate acetaldehyde (*n* = 3/4). **J–L** Phenotypic assessment of *aldh3b1⁻/⁻* adult zebrafish revealed no significant changes in survival rate (**J**, *n* = 50/50), weight (**K**, *n* = 11/11), and length (**L**, *n* = 11/11) compared to *aldh3b1⁺/⁺* adult zebrafish. Dpf, days post fertilization; bp, base pair. Statistical analysis was performed using one-way ANOVA for panels (**D, E**), two-tailed Student's *t* test for (**I, K, L**) and Log-rank test for panel (**J**). Error bars represent mean ± standard deviation (SD). In Fig. 1K, L, the center line represents the median, the bounds of the box represent the 25th and 75th percentiles, and the whiskers extend to the minimum and maximum values. Source data are provided as a Source Data file.

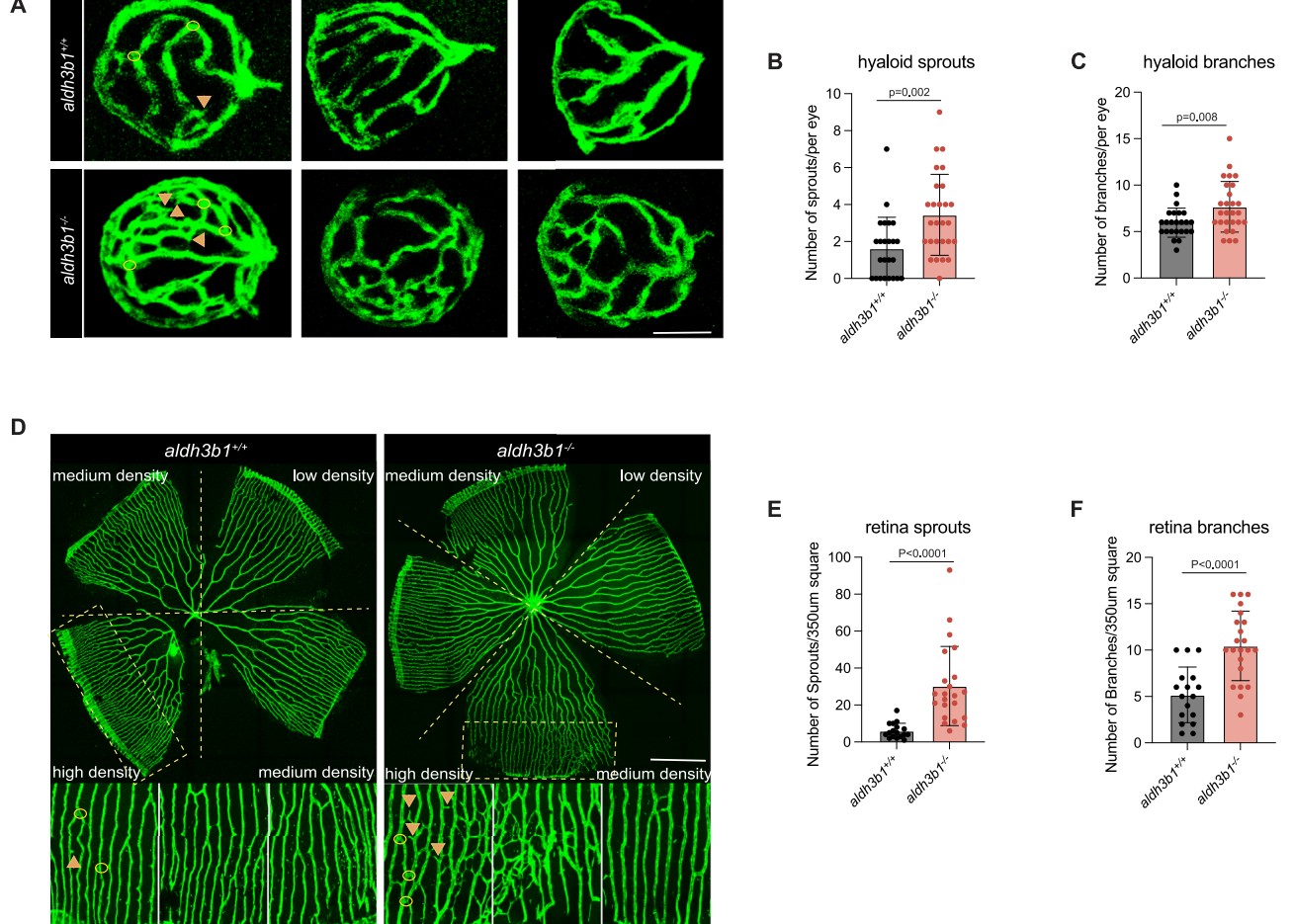

**Fig. 2 | Microvascular alterations in *aldh3b1⁻/⁻* mutants. A** Representative confocal images of hyaloid vasculature revealed significant vascular alterations in *aldh3b1⁻/⁻* larvae at 5dpf. Yellow arrows, sprouts; yellow circles, branchpoints. White scale bar = 50 μm. **B, C** Quantitative analysis demonstrated a significant increase in hyaloid sprouts and branchpoints in *aldh3b1⁻/⁻* larvae (*n* = 24/27). **D** Representative confocal images of adult retinal vasculature showed vascular alterations in

*aldh3b1⁻/⁻* zebrafish. Yellow arrows, sprouts; yellow circles, branchpoints. White scale bar = 500 μm. **E, F** Quantification of increased retinal sprouts and branchpoints in adult *aldh3b1⁻/⁻* zebrafish. One datapoint means one 350 μm² square in high-density retina (*n* = 8/9). The bars indicate mean ± SD values. Statistical analysis was performed by a two-tailed Student's *t* test. Source data are provided as a Source Data file.

## Impaired 2-HD detoxification in *aldh3b1⁻/⁻* mutants altered hyaloid vasculature

2-HD was primarily detoxified by ALDH3B1[21], which was also capable of metabolizing other aldehydes[25]. Therefore, the analysis of other key aldehydes, known to be substrates for ALDH3B1, was performed in *aldh3b1⁻/⁻* mutants to support the use of an *aldh3b1⁻/⁻* knockout model to study 2-HD accumulation effects. The absence of Aldh3b1 was found to have no effect on the metabolism of 4-Hydroxynonenal (4-HNE) and 4-Hydroxyhexenal (4-HHE). However, it significantly reduced NADP+ production in response to tt-DDE and 2-HD treatment, with 2-HD

substrate leading to the strongest reduction in total ALDH enzyme activity (Fig. 3A–D).

To elucidate the effect of 2-HD accumulation on the vascular phenotype, *aldh3b1⁺/⁺* larvae were exposed to different concentrations of 2-HD. Survival rates, cardiac edema, and morphological changes indicated that concentrations above 200 μM caused higher lethality, establishing a range of 0 - 100 μM as safe for further studies (Fig. 3E–G). Incubation with 100 μM 2-HD showed increased hyaloid sprouts and branches, mirroring those observed in *aldh3b1⁻/⁻* mutants (Fig. 3H, I). Importantly, the reactive carbonyl species (RCS) scavenger

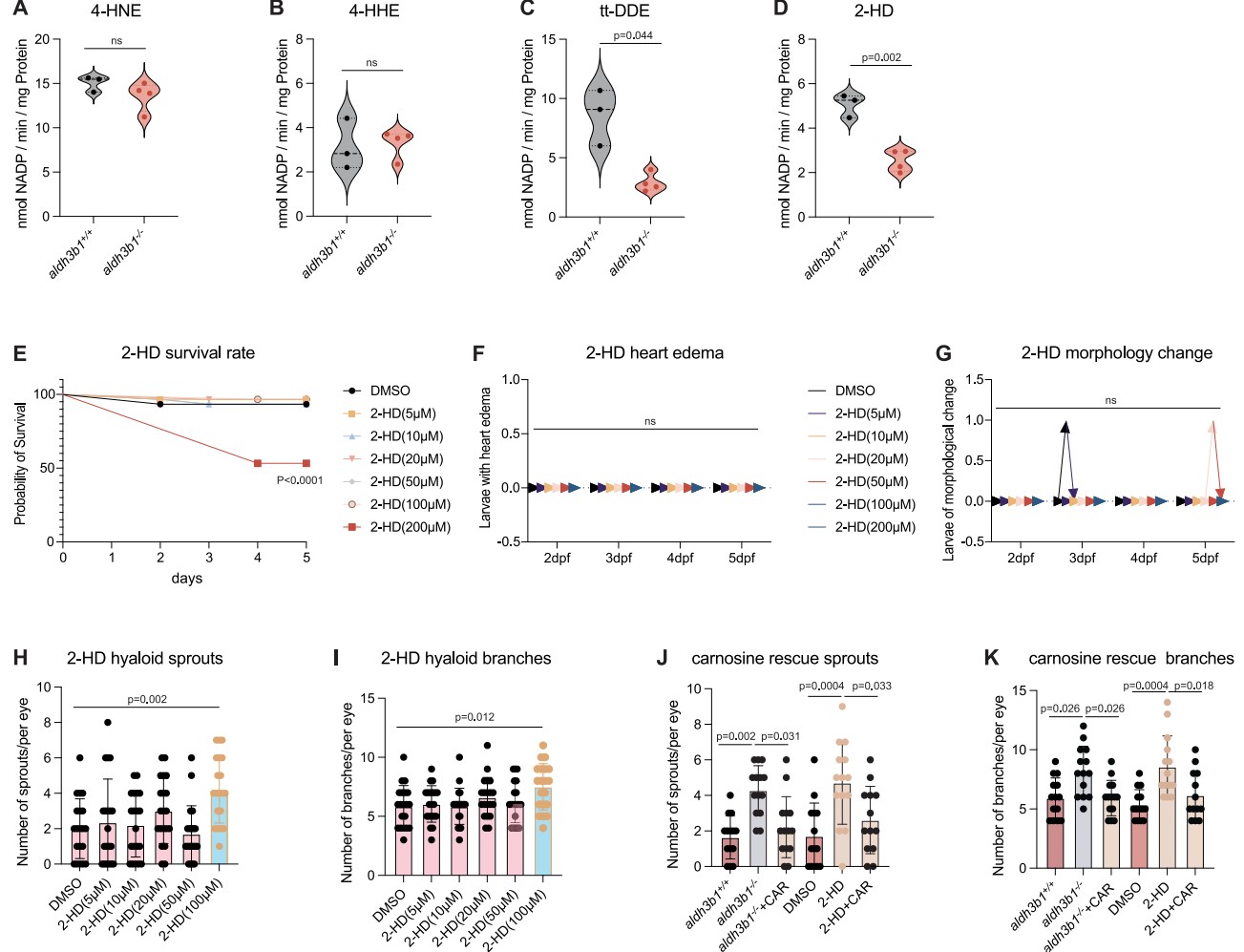

**Fig. 3 | Impaired 2-HD detoxification in *aldh3b1⁻/⁻* mutants altered hyaloid vasculature.** **A–D** ALDH enzyme activity was significantly decreased using substrate tt-DDE (**C**) and 2-HD (**D**), but unaltered with substrate 4-HNE (**A**) and 4-HHE (**B**) in *aldh3b1⁻/⁻* larvae (n = 3/4). **E** Quantification of survival rates using different 2-HD concentrations showed lethality of 200 μmol 2-HD treated zebrafish larvae (n = 3 for each concentration). **F** Quantification of heart edema showed normal larvae heart development within 200 μmol 2-HD (n = 3 for each concentration). **G** Quantification showed normal larvae morphology within 200 μmol 2-HD (n = 3 for each concentration). (**H–I**) 100 μmol 2-HD treatment significantly increased hyaloid sprouts (**H**) and branchpoints (**I**) in *aldh3b1⁺/⁺* larvae (n = 20 for each concentration). **J, K** *aldh3b1⁻/⁻* and 2-HD induced hyaloid vascular alterations can be rescued by carnosine treatment (n = 13 for larvae co-cultured with 2-HD and carnosine, and n = 14 for all other groups). 4-HNE, 4-hydroxynonenal; 4HHE, 4-hydroxyhexenal; tt-DDE, trans, trans-2,4-decadienal; 2-HD, 2-hexadecenal; CAR, carnosine. Statistical analysis was performed using two-tailed Student's *t* test for panels (**A–D**), Log-rank test for (**E**), two-way ANOVA for (**F, G**) and one-way ANOVA for (**H–K**). The bars indicate mean ± SD values. Source data are provided as a Source Data file.

carnosine[26], was effective in reversing the hyaloid phenotypes induced by 2-HD exposure and *aldh3b1* knockout (Fig. 3J, K).

In summary, the comprehensive biochemical and biological analyses have established the *aldh3b1* knockout zebrafish model as an optimal tool to study the in vivo effects of 2-HD accumulation. It was demonstrated that the significant vascular abnormalities observed in *aldh3b1⁻/⁻* mutants are directly attributable to impaired 2-HD detoxification, a consequence of Aldh3b1 deficiency. These findings advanced the understanding of not only 2-HD metabolism but also its effect on vascular phenotypes.

### Activated ferroptosis caused hyaloid vascular abnormalities in *aldh3b1⁻/⁻* and in 2-HD treated zebrafish

To find the underlying mechanism behind the *aldh3b1* knock out and in 2-HD induced vascular alterations, integrative transcriptomic and metabolomic analysis was employed in *aldh3b1⁻/⁻* and in 2-HD treated larvae compared to *aldh3b1⁺/⁺*. Principal component analysis (PCA) of the transcriptomic data displayed distinct clustering of samples within each experimental group (Supplementary Fig. S2A), and 565 and 1659 differentially expressed genes were identified in *aldh3b1⁻/⁻* and in 2-HD treated larvae, respectively. Notably, over 72% of the differential genes in *aldh3b1⁻/⁻* larvae overlapped with those in the 2-HD treated group, reinforcing the relevance of the *aldh3b1⁻/⁻* model for studying 2-HD accumulation effects (Supplementary Fig. S2B). Further analysis by Gene Ontology (GO) and Kyoto Encyclopedia of Genes and Genomes (KEGG) revealed disruptions in iron ion homeostasis and the ferroptosis pathway, respectively, indicating a mechanism of iron dysregulation (Fig. 4A, B and Supplementary Fig. S2C, D). Partial least squares-discriminant analysis (PLS-DA) exhibited the separate groups of metabolomics (Supplementary Fig. S2E). A combined transcriptomic and metabolomic investigation illuminated significant alterations in core ferroptosis pathways, in particular the inducer-iron metabolism and the suppressor Glutathione Peroxidase 4 (GPX4) system[27] (Fig. 4C). The interplay between transferrin receptor (*tfr*) upregulation, enhancing cellular iron uptake, and *gpx4* downregulation, facilitating lipid peroxidation, confirmed ferroptosis activation[27] (Fig. 4C). Parallel metabolic and oxidative stress markers were identified in the eyes, mirroring the larval observations (Supplementary Fig. S2F, G).

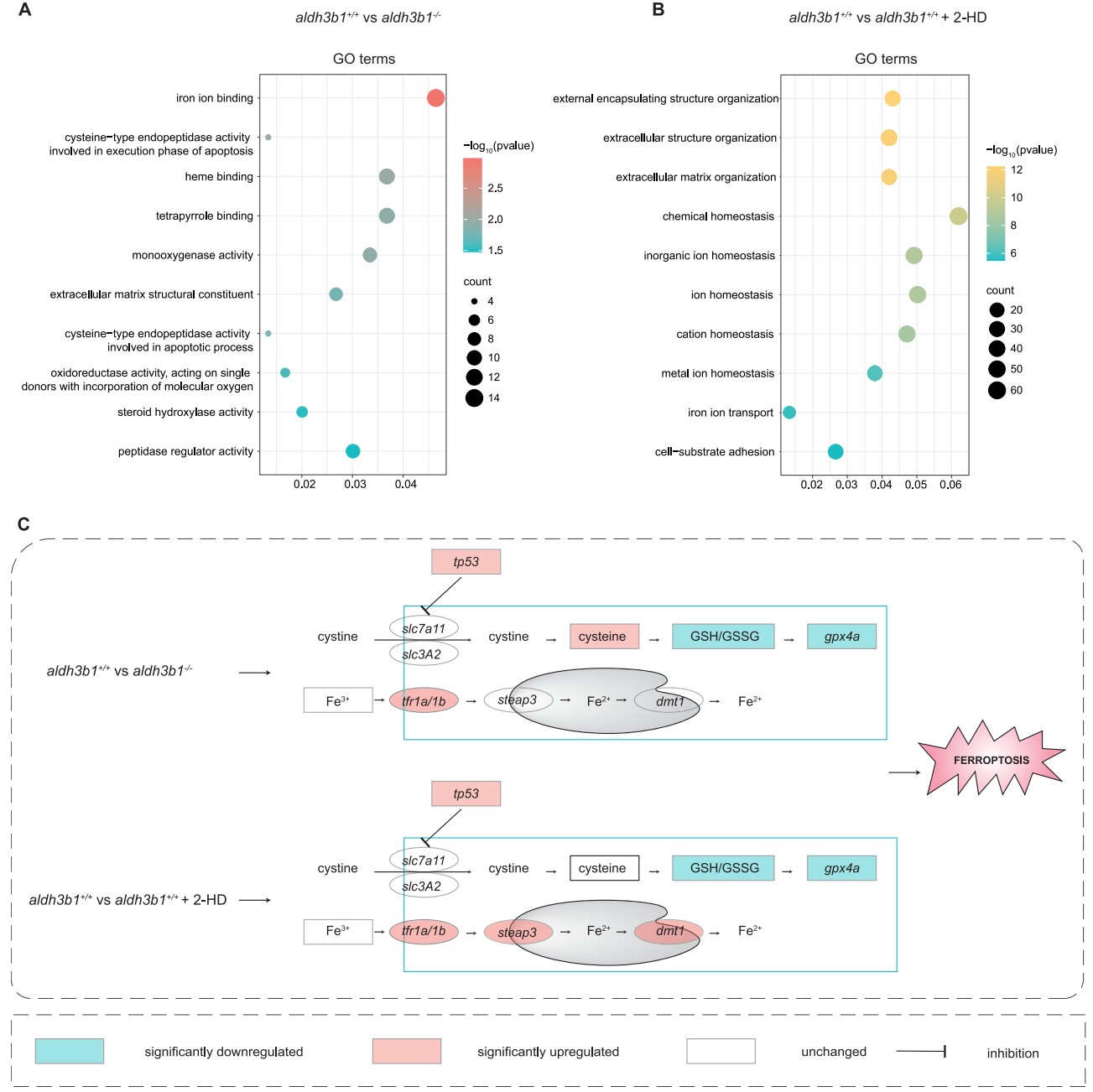

**Fig. 4 | Alterations of ferroptosis related pathways in *aldh3b1^(-/-)* and in 2-HD treated zebrafish. A** GO enrichment analysis of differential genes showed top altered pathways in *aldh3b1^(-/-)* compared to *aldh3b1^(+/+)* larvae. **B** GO enrichment analysis of differential genes showed the top altered pathways in *aldh3b1^(+/+)* 5dpf larvae treated with or without 2-HD. **C** Integrative transcriptome and metabolome analysis uncovered ferroptosis signaling alterations induced by *aldh3b1* deficiency and 2-HD treatment. GO, Gene Ontology. Statistical significance for GO enrichment in Fig. 4A, B was determined using a two-sided Fisher's exact test. P values were adjusted for multiple comparisons using the Benjamini-Hochberg method.

To validate activated ferroptosis and find its key regulator, an in-depth analysis of its biomarkers and seven canonical pathways was conducted[27] (Fig. 5A). Key ferroptosis markers, lipid peroxidation, and iron accumulation were detected in both larvae and eyes. Increased levels of malondialdehyde (MDA), a primary biomarker of lipid peroxidation[28], were observed in *aldh3b1^(-/-)* and 2-HD-treated larvae and eyes (Fig. 5B, C). Due to analytical constraints, the increase in iron ions was only quantifiable in larvae (Fig. 5D). Screening of ferroptosis-associated pathways (Supplementary Fig. S3A, B) revealed a significant downregulation of *gpx4a* and ferroptosis Suppressor Protein 1 (*fsp1*), in *aldh3b1^(-/-)* larvae, and a similar trend, though not statistically significant for *gpx4a*, in 2-HD−treated larvae, at both gene (Fig. 5E, F) and

protein levels (Fig. 5G), with consistent findings in adult eyes (Fig. 5H−K).

Given the consistent decrease in Fsp1 across models, it was hypothesized to be a key regulator of ferroptosis. Validation experiments employing the Fsp1 inhibitor iFSP1−which did not adversely affect survival, morphology, or heart development at concentrations of 0−20 μM (Supplementary Fig. S3C-D) - mirrored the hyaloid vascular phenotypes observed in *aldh3b1^(-/-)* and in 2-HD treated larvae (Fig. 5L, M), and successfully decreased Fsp1 protein levels (Supplementary Fig. S3F). In addition, treatment with the ferroptosis inhibitor Ferr-1[29,30] partially rescued the hyaloid vascular abnormalities in *aldh3b1^(-/-)* and in 2-HD treated larvae, although the effect on vascular

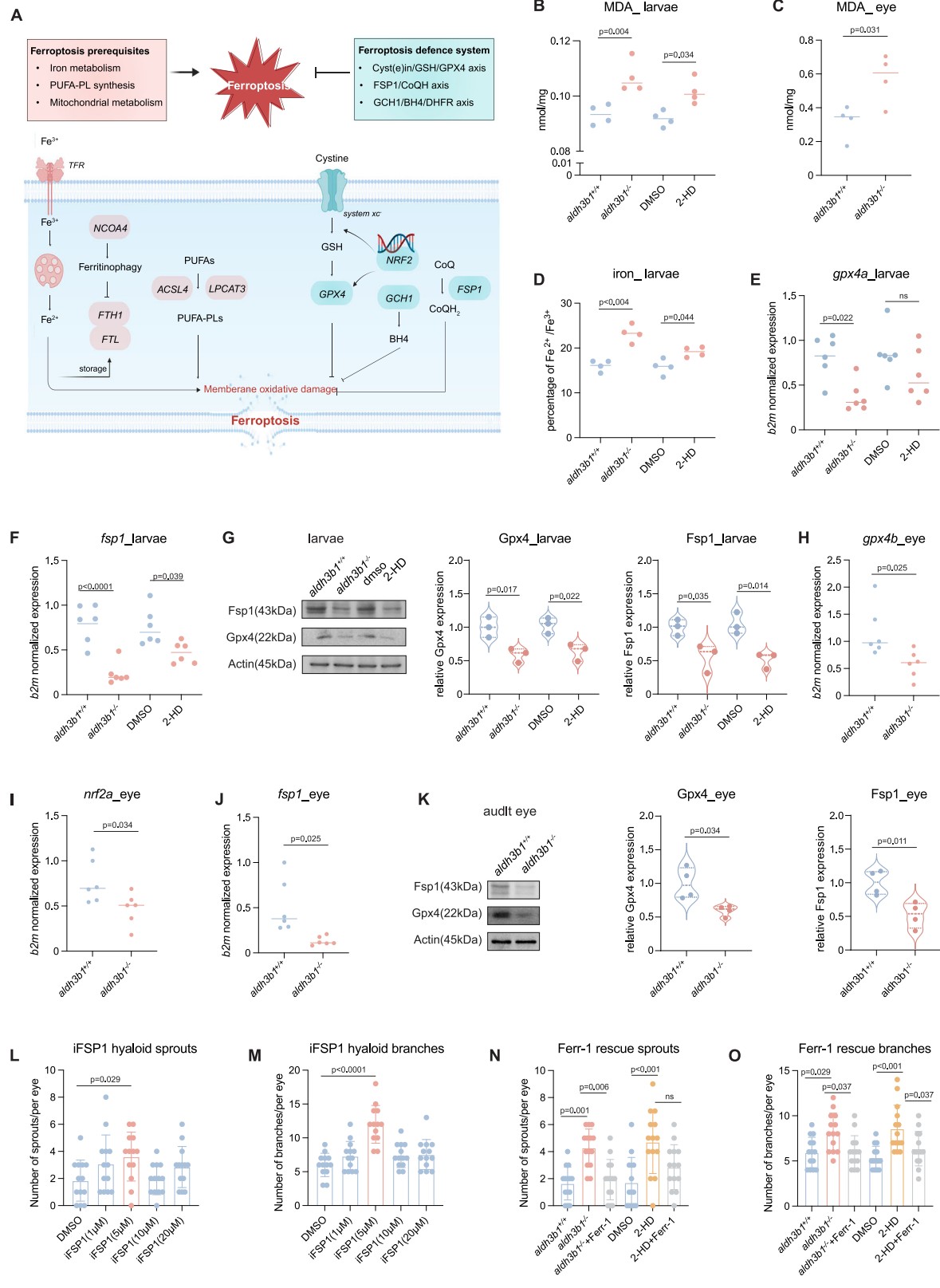

sprouts in 2-HD–treated larvae did not reach statistical significance (Fig. 5N, O), conclusively implicating ferroptosis as the critical mechanism behind these phenotypes in both *aldh3b1⁻/⁻* and in 2-HD treated zebrafish.

## S1pr5 regulates Fsp1 and hyaloid vasculature

Sphingosine-1-Phosphate regulates cell survival, vascular homeostasis, and tissue modeling via its receptors S1PR 1-5[31], and 2-HD is one of its degradation product[12,32]. Therefore, we hypothesized that altering

**Fig. 5 | Activated ferroptosis caused hyaloid vascular abnormalities in *aldh3b1*[-/-]** **and in 2-HD treated zebrafish. A** Graphical view of the occurrence and regulatory pathways involved in ferroptosis. Created in BioRender. Bennewitz, K. (2026) https://BioRender.com/p96mk4e. **B, C** *aldh3b1* deficiency and 2-HD treatment increased lipid peroxidation as determined by increased MDA (malondialdehyde) formation in zebrafish larvae (**B**, *n* = 4 for each group) and eye (**C**) (*n* = 4/4). **D** *aldh3b1* deficiency and 2-HD treatment increased the percentage of $Fe^{2+}$/$Fe^{3+}$ in larvae (*n* = 4 for each group). **E, F** Reduced *gpx4* and *fsp1* mRNA expression caused by *aldh3b1* deficiency and 2-HD treatment (*n* = 6 for each group). **G** Immunoblot analysis revealed decreased Fsp1 and Gpx4 proteins in *aldh3b1*[-/-] and 2-HD treated *aldh3b1*[+/+] larvae (*n* = 3 for each group). **H–J** Reduced *fsp1*, *gpx4* and *nrf2a* mRNA

levels in eyes from adult *aldh3b1*[-/-] zebrafish (*n* = 6/6). **K** *aldh3b1* deficiency decreased Fsp1 and Gpx4 proteins in the zebrafish eye (*54* = 4/4). **L, M** 10 μmol iFSP1 significantly increased hyaloid sprouts (**L**) and branchpoints (**M**) in *aldh3b1*[+/+] larvae (*n* = 13 for each concentration). **N, O** *aldh3b1*[-/-] and 2-HD induced hyaloid vascular alterations can be rescued by Ferrostatin-1 (*n* = 14 for each group). MDA, malondialdehyde; GPX4, glutathione peroxidase 4; FSP1, ferroptosis suppressor protein 1; NRF, nuclear factor erythroid 2-related factor; Ferr-1, ferrostatin-1. Statistical analysis was performed using two-tailed Student's *t* test for panels (**C, H–K**) and one-way ANOVA for (**B, D–G** and **L–O**). The bars indicate mean ± SD values. Source data are provided as a Source Data file.

2-HD levels by manipulating *aldh3b1* would influence vascular development and ferroptosis via S1P signaling pathways. Gene set enrichment analysis (GSEA) demonstrated significant alterations in S1P-related pathways, particularly sphingolipid metabolism, in *aldh3b1*[-/-] and in 2-HD treated larvae (Fig. 6A). Examination of the five S1P receptors (Supplementary Fig. S4A) indicated a marked reduction in S1pr5, including both *s1pr5a* and *s1pr5b* isoforms, at both RNA (Supplementary Fig. S4A) and protein levels (Fig. 6B). This reduction was similarly observed in adult zebrafish eyes (Supplementary Fig. S4B, 6C).

To further validate our hypothesis, morpholinos (MOs) targeting *s1pr5a* and *s1pr5b* were designed to knock down the S1pr5 to assess its impact on the hyaloid vasculature. The MOs didn't affect survival rate, morphology and heart development (Supplementary Fig. S4C, E), but induced hyaloid vascular abnormalities comparable to those found in *aldh3b1*[-/-] and in 2-HD treated models (Fig. 6D, E). Immunoblotting confirmed the effective knockdown of S1pr5 and also noted a decrease in Fsp1 levels, with no significant impact on Gpx4 levels (Fig. 6F). Utilization of A-971432 (A-971), a selective S1PR5 agonist[33], further examined the role of S1pr5 in the hyaloid phenotypes. An 8 μM concentration of A-971 was identified to be safe (Supplementary Fig. S4F–H) and optimal (Supplementary Fig. S4I, J) for rescuing the phenotypes in 2-HD-treated larvae and was also found to be effective in *aldh3b1* knockout models (Fig. 6G-H).

Since S1P is a primary regulator of S1PR5, the association between 2-HD and S1PR5 was thought to be due to an imbalance in S1P. However, S1P levels remained unchanged in both larvae and eyes (Supplementary Fig. S4K, L).

To explore a potential interaction between 2-HD and the S1PR5, both binding and receptor internalization were examined. Surface Plasmon Resonance (SPR) binding assay demonstrated that 2-HD binds to S1PR5 in a concentration-dependent manner (Figs. S4M and 6I). In primary human natural killer (NK) cells, treatment with 100 or 200 μmol 2-HD reduced surface S1PR5 signal (Figs. S4N and 6J), without altering total S1PR5 protein levels (Supplementary Fig. S4O), indicating that 2-HD induces receptor internalization. To further characterize this interaction, docking analysis was performed. Alpha, beta-unsaturated fatty acids aldehyde, such as 2-HD, has been reported to modify the protein by Michael addition reaction, particularly targeting cysteine residues[34]. Our computational model identified the specific adduct between 2-HD and S1PR5 that shared similar binding sites with the positive control ONO-5430608 (ONO)-S1PR5[35] (Fig. 6K). Further comparative analysis, including an S1P-S1PR3 complex, demonstrated a consensus in binding sites among ONO, S1P, and 2-HD, thereby suggesting the successful establishment of a 2-HD and S1PR5 interaction model (Fig. 6L). These findings indicate that 2-HD is functionally linked to S1P signaling and associated vascular phenotypes, with S1PR5 emerging as a key downstream mediator.

## S1PR5 is a novel target for human diabetic retinopathy

To determine the applicability of our findings to humans, we conducted a comprehensive analysis incorporating single-cell RNA sequencing (scRNA-seq) and RNA-seq data derived from human retinal

samples, which also included choroidal tissue. This scRNA-seq analysis integrated data from the healthy adult retinas from two independent studies[36,37], resulting in a dataset of 67,208 cells. By utilizing Uniform Manifold Approximation and Projection (UMAP) for clustering, 15 typical retinal cell-types with specific markers were identified, as previously reported[36,37] and detailed in our methods section (Fig. 7A). Among these, three genes of interest—*ALDH3B1*, *S1PR5*, and *FSP1* - known to be critical in zebrafish retinal neovascularization, showed significant expression in microglia (MG), NK cells, and fibroblasts, respectively (Fig. 7B–D and Supplementary Fig. S5A). Notably, *S1PR5* showed a more specific distribution pattern compared to *ALDH3B1* and *FSP1*.

While *aldh3b1* has been found in diabetes models, its absence has been shown not to correlate with alterations in glucose metabolism, suggesting that retinal neovascularization is the predominant phenotype in *aldh3b1*[-/-] zebrafish. Consequently, we examined three human retinal diseases that are mainly characterized by retinal neovascularization - AMD, branch retinal vein occlusion (BRVO), and DR - to assess the role of these genes in human retinal neovascularization. Analysis of RNA-seq data from patients with AMD[38], BRVO[39] and DR[39–41], obtained from the Gene Expression Omnibus (GEO) database, revealed no significant changes in the expression levels of the three genes in AMD and BRVO (Fig. 7C, D and Supplementary Fig. S5B, C). However, in proliferative diabetic retinopathy (PDR), *S1PR5* expression levels were significantly elevated in proliferative stages, whereas *ALDH3B1* and *FSP1* remained unchanged (Fig. 7E and Supplementary Fig. S5B, C). GSEA analysis further differentiated the pathogenesis of these diseases based on KEGG categories of significant pathways, providing a potential explanation for the expression patterns of *ALDH3B1*, *S1PR5*, and *FSP1* in these diseases, particularly in PDR (Fig. 7F–H).

In addition, an analysis of S1P-related signaling in human PDR highlighted notable alterations in sphingolipid, extracellular matrix (ECM), and T/NK cell signaling pathways, mirroring the pathway changes observed in *aldh3b1* knockout and 2-HD treated zebrafish larvae (Fig. 7I). Correlation analyses of *S1PR5* with other genes identified similar pathway modifications to those seen in PDR, establishing S1PR5 as a key regulator in the pathogenesis of PDR (Fig. 7J). Further investigation confirmed S1PR5's association with ferroptosis in human diabetes (Fig. 7K).

To validate these transcriptomic findings, a diabetic zebrafish model (*pdx1*[-/-]) was examined. S1pr5 expression in *pdx1*[-/-] larvae mirrored that observed in human PDR samples (Supplementary Fig. S5D), and the vascular abnormalities in the *pdx1*[-/-] hyaloid vasculature were rescued by ferroptosis inhibition or S1PR5 activation (Supplementary Fig. S5E–H). To further explore the cellular mechanisms, co-culture experiments of human microglia–NK cells, NK–fibroblasts, and NK–endothelial cells (ECs) were performed, guided by the scRNA-seq results. Conditioned medium from ALDH3B1-deficient microglia reduced S1PR5 expression in NK cells (Supplementary Fig. S5I, J). Moreover, S1PR5 knockdown in NK cells increased MDA and iron accumulation while reducing FSP1 expression in fibroblasts and endothelial cells (Supplementary Fig. S5K–Q).

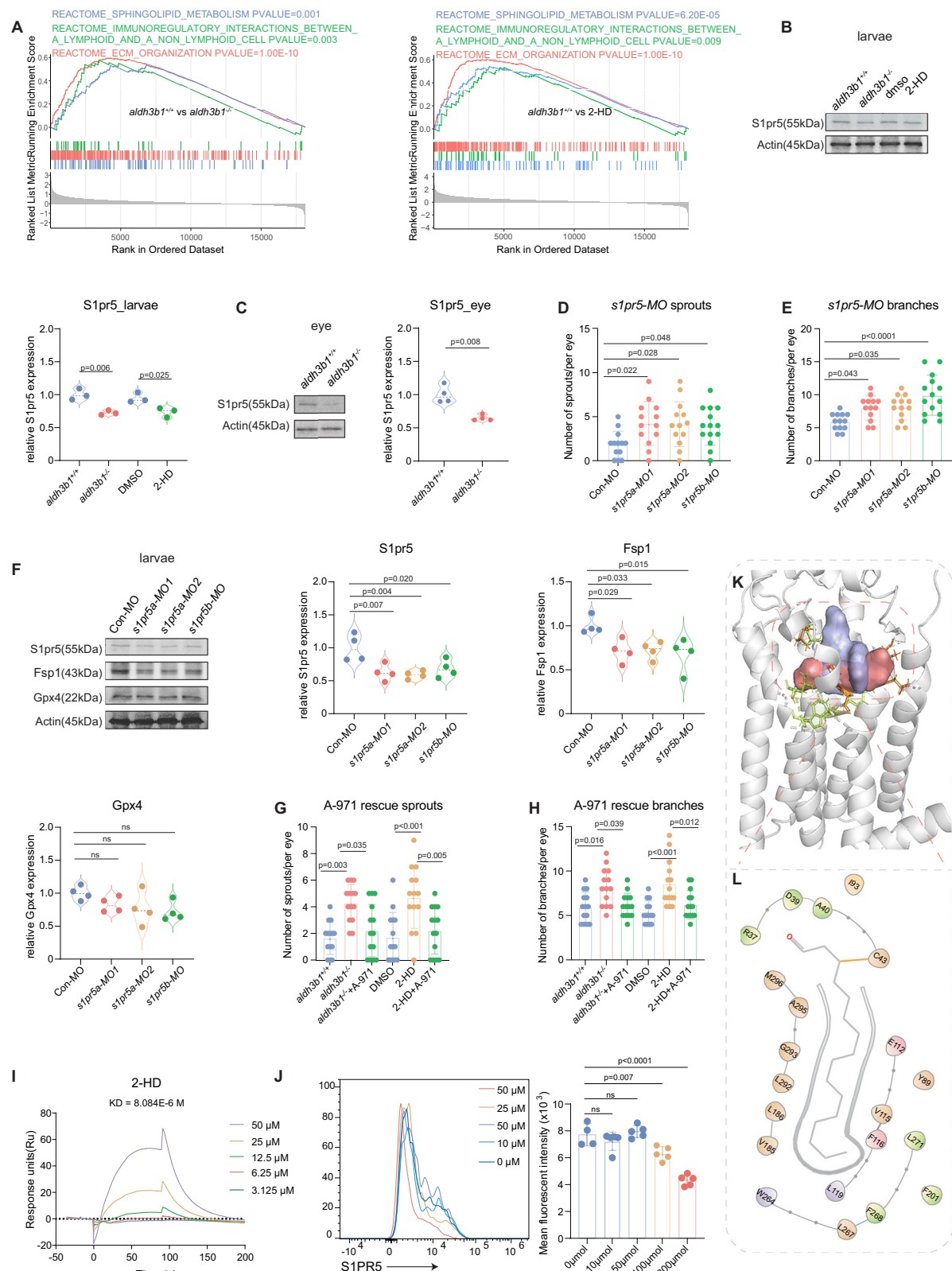

These results corroborate our zebrafish observations and establish S1PR5 as a novel therapeutic target, highlighting its critical role in retinal neovascularization and the regulation of ferroptosis in human PDR.

## Discussion

Sphingosine-1-phosphate (S1P) is known for its essential role in angiogenesis and the maintenance of vascular integrity, whereas the vascular effects of 2-HD, a major degradation product of S1P, remain unexplored. In this study, we established an *aldh3b1*[-/-] zebrafish model to investigate the accumulation of 2-HD and its vascular effects. Vascular analysis of both macrovasculature and microvasculature revealed increased pathological neovascularization in the retinas of *aldh3b1*[-/-] mutants and 2-HD-treated zebrafish. Mechanistically, S1PR5-related ferroptosis emerged as a significant pathway mediating the

**Fig. 6 | S1pr5 regulates Fsp1 and hyaloid vasculature in *aldh3b1⁻ᐟ⁻* and in 2-HD treated zebrafish. A** GSEA analysis of transcriptome displayed significant change of sphingolipid metabolism and related pathway in *aldh3b1⁻ᐟ⁻* and 2-HD treated *aldh3b1⁺ᐟ⁺*larvae. **B** *aldh3b1* deficiency and 2-HD treatment decreased S1pr5 protein in zebrafish larvae ($n = 3$ for each group). **C** *aldh3b1* deficiency decreased S1pr5 protein in the zebrafish eye ($n = 4/4$). **D, E** Reduced S1pr5 expression by *s1pr5a/b* morpholinos (MO) injection increased hyaloid sprouts (**D**) and branchpoints (**E**) of 5dpf zebrafish larvae ($n = 14$ for larvae injected with *s1pr5b* MO, and $n = 13$ for all other groups). **F** S1pr5 morpholinos injection decreased S1pr5 and Fsp1 proteins, but Gpx4 remained unaltered ($n = 4$ for each group). **G, H** *aldh3b1⁻ᐟ⁻* and 2-HD induced hyaloid vascular alterations can be rescued by the S1PR5 agonist A-971432 ($n = 14$ for each group). **I** SPR assay revealed 2-HD bind to S1PR5 in a concentration-dependent manner. **J** Flow cytometry analysis showed 100 and 200 µmol 2-HD significantly decreased surface S1PR5 signal of NK cells ($n = 5$). **K** Computational model demonstrated interaction modes between 2-HD and S1PR5 with ONO-5430608-S1PR5 as controls. The red surface highlights 2-HD, the purple surface denotes ONO-5430608 (ONO), yellow sticks represent consensus residue shared by two modulators, and green sticks indicate 2-HD unique binding sites. (**L**) Schematic diagram of the ligand binding pocket and interactions between 2-HD and S1PR5 compared to interactions of ONO-S1PR5 and S1P-S1PR3. Residues are color-coded according to their interaction specificity. Yellow for consensus residues between 2-HD and ONO, green for 2-HD's unique binding sites, purple for consensus residues shared between 2-HD and S1P, and pink for consensus residues shared among ONO, 2-HD, and S1P. ECM, extracellular matrix; S1PR5, Sphingosine-1-phosphate receptor 5; MO, morpholino; A-971, A-971432; SPR, Surface Plasmon Resonance. Statistical analysis was performed using one-way ANOVA for panels (**B, D–H**, and **J**) two-tailed Student's *t* test for C. The bars indicate mean ± SD values. For Fig. 6A, GSEA was performed using a permutation-based statistical test. Significance was assessed using the Normalized Enrichment Score (NES) and the False Discovery Rate (FDR) *q*-value to account for multiple hypothesis testing. Source data are provided as a Source Data file.

vascular pathology associated with 2-HD exposure. To strengthen these findings, human retinal tissue analysis validated that S1PR5 was associated with diabetic retinopathy more significantly than in other ocular neovascular diseases. This suggests S1PR5 modulation as a potential therapeutic target for managing DR.

Our previous research has highlighted the importance of α,β-unsaturated aldehydes such as acetaldehyde[42], 4-HNE[18], and tt-DDE[19], in diabetes and DR. In diabetes, hyperglycemia and a stressful environment increase oxidative stress, which catalyzes the oxidation of carbohydrates, lipids, and amino acids, leading to the formation of these aldehydes[43,44]. Like other α,β-unsaturated aldehydes, 2-HD can also be generated from the free radical-mediated degradation of S1P[10]. Moreover, S1P levels have been found to be elevated in the serum[45] and aqueous humor[46] of diabetic patients and in the mouse retina under hypoxic or light-induced stress[47,48], highlighting the potential importance of 2-HD accumulation in DR.

The currently known role of 2-HD has primarily been associated with cell growth and death[11–15]. Specifically, 2-HD can activate BAX (BCL-2-associated X protein) through covalent derivatization of cysteine 126 (C126) to induce apoptosis[49] or modulate NADPH oxidase and MAPK pathways to generate reactive oxygen species (ROS)[15], in contrast to the inhibitory role of S1P on cell death[12,31]. Based on these observations, we hypothesized that 2-HD, as a downstream metabolite of S1P metabolism, may act antagonistically to S1P, particularly in the key processes of angiogenesis and cell survival. Our findings validated this hypothesis, showing that 2-HD induced abnormal retinal angiogenesis, whereas S1P is typically considered protective in DR. These findings are consistent with our previous data and those of others demonstrating the significant effect of various reactive carbonyl species (RCS)-including 4-HNE, tt-DDE, acrolein, and acetaldehyde- on retinal angiogenesis[18,19,42]. The inclusion of 2-HD as a newly identified RCS expands our understanding of how specific RCS profiles contribute to pathological retinal angiogenesis.

Regarding cell death, we identified that ferroptosis, in contrast to apoptosis, is a key mechanism in the neovascular retina. This may be due to the fact that previous studies have mainly focused on in vitro tissue culture models without considering retina-specific cells and the unique structural characteristics of the retina[50,51]. The retina, rich in polyunsaturated fatty acids and the most oxygen-consuming tissue in the body, is constantly exposed to light, making it exceptionally susceptible to peroxidative damage from ROS[52]. In addition, the importance of iron balance in the retina is highlighted by the presence of iron-rich proteins such as neuroglobin (Ngb) for oxygen transport, the retinal pigment epithelium-derived 65 kDa protein for rhodopsin production, and the heme-containing guanylate cyclase involved in the phototransduction cascade[52]. These two features are fundamental prerequisites for ferroptosis[27] and provide a plausible explanation for the predominance of this form of cell death in the retina. In addition,

our analysis of scRNA-seq data from human retinas has shown that *FSP1* is mainly located in fibroblasts, which are prominent in the late stages of DR, characterized by the formation of fibrovascular membranes (FVMs). FVMs are essentially scar tissue that forms due to abnormal growth of new blood vessels in the retina, the hallmarker of end-stage DR and have several harmful effects, including vitreous hemorrhage and tractional retinal detachment. Therefore, we postulate that ferroptosis may represent a late-stage mechanism in the progression of retinal neovascularization.

Considering the contrasting functions of 2-HD and S1P, and the mediation of S1P biological functions by its receptors, we examined the alterations in S1P levels and S1PRs expression in the *aldh3b1⁻ᐟ⁻* zebrafish. Intriguingly, only *s1pr5* showed significant changes, while S1P and other receptors remained unchanged. S1PR5 has been less investigated in previous studies compared to other receptors, with current knowledge focusing mainly on immune regulation[53–59]. S1PR5 is primarily found in CD8⁺ T and NK cells, as well as in the brain[53,54,56,59]. Our sc-seq analysis of the human retina mirrored previous findings and also identified *S1PR5* predominantly in NK cells, where it functions similarly to S1PR1 in regulating NK cell trafficking and emigration from peripheral organs in both the innate and adaptive immune compartments[53,54,56,59]. Loss of S1PR5 enhances the formation of tissue-resident memory NK cells, attenuates local inflammation, and can induce autoimmune diseases[54], which is a central mechanism for the use of approved S1PR-related drugs in MS patients[60]. In addition, a clinical trial demonstrated a significant correlation between S1PR5 levels and diabetes incidence in patients[57] and many S1PR5 modulators like ONO have been shown to delay the onset of diabetes in animal models[61,62]. This is consistent with our data suggesting a significant upregulation of *S1PR5* expression in PDR, a disease predominantly characterized by immune dysregulation compared to other ocular neovascular diseases. Growing evidence supports targeting immune regulation as a strategic approach for the treatment of PDR[63,64]. Besides, a clinical study found that S1P levels were significantly elevated only in the aqueous humor of patients with PDR and not in non-proliferative diabetic retinopathy (NPDR) patients[46], explaining our observations of increased S1PR5 in PDR cases only. While our biochemical assays and computational modeling are consistent with a potential interaction between 2-HD and S1PR5, these approaches have important limitations. The micromolar binding affinities observed in the SPR experiments, including for the positive control siponimod, indicate that assay conditions or receptor conformation may limit definitive conclusions about direct binding. In addition, given the electrophilic nature of 2-HD, receptor-independent mechanisms such as covalent modification of other hydrophobic proteins in the endoplasmic reticulum or nucleus cannot be excluded. Finally, our binding data do not establish receptor selectivity. Therefore, although S1PR5 is functionally implicated in the observed phenotype, the precise

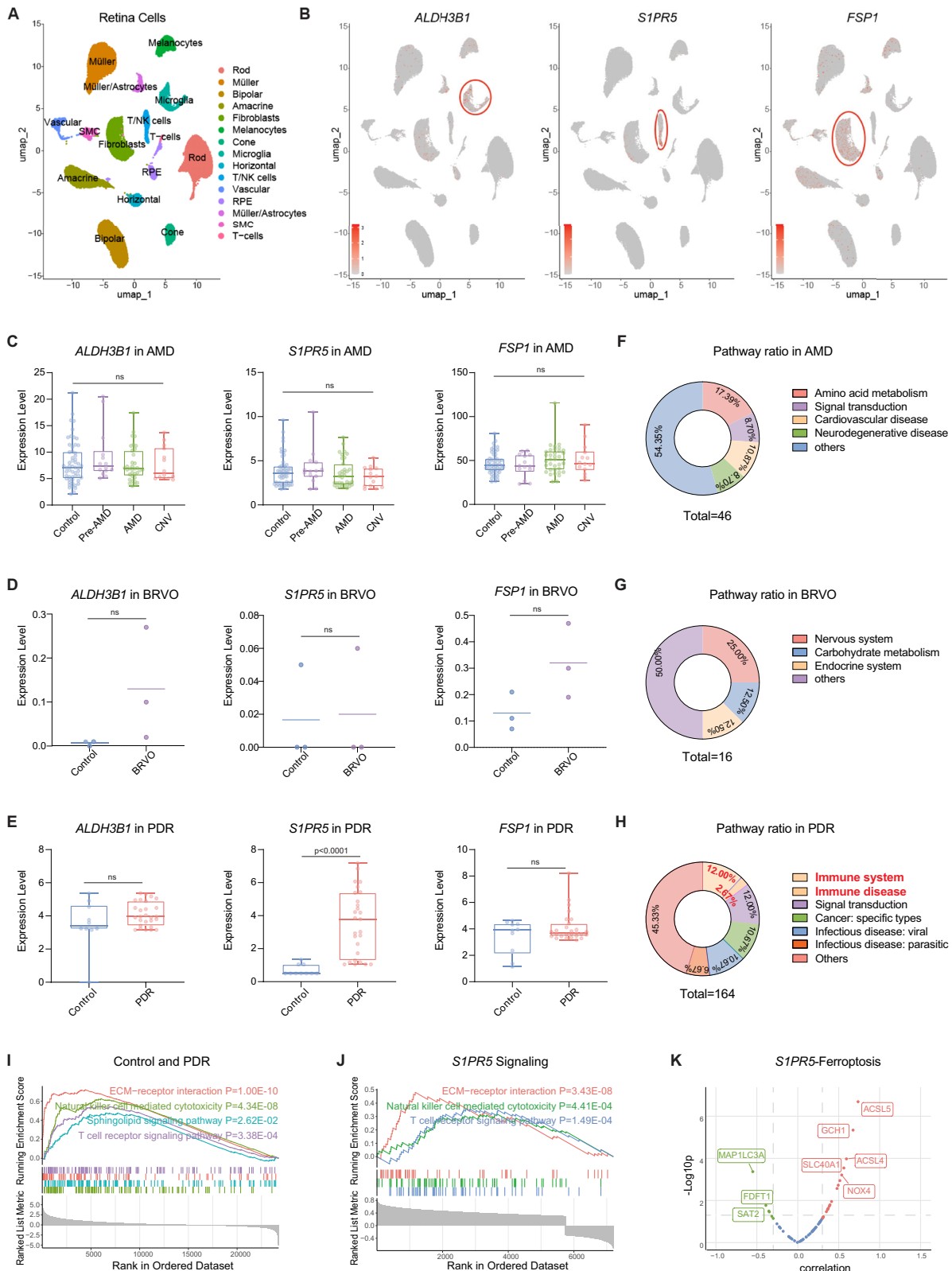

molecular mechanism of 2-HD-S1PR5 binding remains to be clarified. These results suggest S1PR5 as an initial driving factor of the observed retinal neovascularization.

Therapeutic strategies targeting retinal neovascularization typically require frequent intravitreal administration and are associated with numerous adverse effects. Recently, a clinical trial demonstrated that Fingolimod, an oral agonist of all S1PRs except S1PR2, is beneficial for ocular diseases in MS patients[7]. Since S1PR1 and S1PR5 are key mechanisms in the treatment of MS patients, with S1PR1 well studied for its protective role in DR and its function similar to that of S1PR5, our studies have characterized the role of S1PR5 in DR for the first time, suggesting its therapeutic value in the management of this disease. As more S1PR modulators emerge, we believe our study will shed new light on treatment options for DR patients and drug development.

**Fig. 7 | S1PR5 is a novel target for human diabetic retinopathy. A** UMAP integrated sc-RNA sequencing identified 15 typical retinal cell clusters. **B** Expression of *ALDH3B1*, *S1PR5*, and *FSP1* was primarily observed in microglia, T/NK cells, and fibroblasts, respectively. **C**–**E** Integrated RNA-seq data showed gene expression variations in retinal diseases with eovascularization, including AMD (*n* = 55/14/35/14), BRVO (*n* = 3/3), and DR (*n* = 10/27). The expression level of three examined genes were not altered in AMD and BRVO. But in PDR patients, *S1PR5* expression was identified to be significantly increased. **F**–**H** KEGG pathway enrichment analysis of differential genes highlighted significant changes in pathway categories. **I** GSEA analysis of transcriptome revealed significant alterations in sphingolipid metabolism and related pathways in PDR retina compared to healthy retina. **J** GSEA of *S1PR5* related genes showed similar changes to those observed in PDR.(**K**) PCC analysis indicated an expression correlation between *S1PR5* expression and ferroptosis-related genes in PDR samples. UMAP, Uniform Manifold Approximation and Projection; sc-RNA, single cell RNA sequence; AMD, age-related macular degeneration;

CNV, choroidal neovascularization; BRVO, branch retinal vein occlusion; DR, diabetic retinopathy; PDR, proliferative diabetic retinopathy; KEGG, Kyoto Encyclopedia of Genes and Genomes; GSEA, Gene set enrichment analysis; PCC, Pearson correlation coefficient. Pathway categories were annotated using the KEGG subcategory. S1PR5-related genes were selected with a threshold of $R > 0.3$ and $p < 0.05$. For Fig. 7C, E, the center line represents the median, the bounds of the box represent the 25th and 75th percentiles, and the whiskers extend to the largest and smallest values within 1.5 times the interquartile range. Individual points beyond the whiskers are plotted as outliers. Statistical comparisons of gene expression in Fig. 7C–E were performed using Student's *t* test. For Fig. 7I, J, GSEA was performed using a permutation-based statistical test. Significance was assessed using the NES and the FDR q-value to account for multiple hypothesis testing. Statistical significance in Fig. 7k was calculated using a two-sided Pearson correlation. Source data are provided as a Source Data file.

In conclusion, the zebrafish model has proven to be useful in providing a previously unexpected explanation for the microvascular disorders seen in DR patients. Our data underscore the crucial role of ALDH3B1 in the control of S1P-derived 2-HD, which, if insufficiently detoxified, triggers the S1PR5-FSP1 axis, leading to microvascular disease.

## Methods

### Zebrafish models
All the zebrafish experimental protocols were approved by the local government authority Regierungsprasidium–Karlsruhe and by Medical Faculty Mannheim (license no: G-98/15 and I-19/02) and carried out in accordance with the approved guidelines.

The zebrafish line *Tg(fli1:EGFP)* was used and raised as established husbandry environment[65,66]. Embryos or larvae were cultured in E3 medium at 28.5 °C for the first 5 days, then they were transferred to adult boxes under a 13 h light/11 h dark cycle. Adult fish were fed twice a day, freshly hatched Artemia Salina in the morning and fish flake food in the afternoon.

### Preparation of NK cells
Peripheral blood samples were obtained from donors in the Department of Vascular Surgery, Renji Hospital, in accordance with a study protocol approved by the Institutional Review Board of Shanghai Jiao Tong University School of Medicine, Renji Hospital (Approval No. KY2024–170-C). All procedures involving human participants adhered to the ethical principles of the Declaration of Helsinki. Written informed consent was obtained from all participants or their legal guardians prior to specimen collection. All enrolled donors were considered as healthy and had no known underlying diseases.

Peripheral blood mononuclear cells (PBMCs) were isolated from 10 mL of venous blood. Briefly, whole blood was diluted 1:1 with phosphate-buffered saline (PBS) and carefully layered onto Ficoll® PM 400 separation medium (F4375, Sigma-Aldrich). Samples were centrifuged at $400 \times g$ for 20 min at room temperature without brake. After centrifugation, the PBMC layer at the plasma–Ficoll interface was carefully collected and washed four times with RPMI-1640 medium to remove residual platelets and Ficoll. PBMCs were then resuspended at $1 \times 10^7$ cells/mL in NK cell culture medium (ALyS505NK-AC, BASO) and incubated at 37 °C in a humidified atmosphere containing 5% $CO_2$.

The PBMCs then were purified by flow cytometry. PBMCs were incubated with Fc receptor–blocking reagent (S0F0008-100T, STARTER) for 10 min at 4 °C and stained for 30 minutes at 4 °C in the dark with monoclonal antibodies against CD3 (FITC, 555339, BD Pharmingen), CD56 (BV421, 562751, BD Pharmingen), and a LIVE/DEAD viability dye (BV510, AB_2869572, BD Horizon™). Cells were washed, passed through a 40-μm strainer, and sorted on a BD FACSAria™ III. NK cells were defined as viable CD3⁺CD56⁺ lymphocytes, and the final purity consistently exceeded 95%.

Purified NK cells were cultured in NK cell medium (ALyS505NK-AC, BASO) at 37 °C in a humidified incubator with 5% $CO_2$ and were activated by supplementing the medium with recombinant human IL-2 (1000 IU/mL) and IL-15 (10 ng/mL). Cells were incubated under these conditions for 24 h before being harvested for downstream assays.

### Cell culture
Human NK cells were cultured in NK cell medium (ALyS505NK-EX, BASO) at 37 °C in a humidified incubator with 5% $CO_2$. Human fibroblasts (HEF, SCSP-106, National Collection of Authenticated Cell Cultures, China) and human microglia (donated by Dr. Yan Li[67]) were maintained in DMEM (Gibco) supplemented with 10% fetal bovine serum (FBS) under the same incubation conditions. Human umbilical vein endothelial cells (HUVECs; DFSC-EC-01, ZQXZbio) were cultured in endothelial cell medium (#1001, ScienceCell, United States). siRNA targeting ALDH3B1 was obtained from Cas9X China, and shRNA targeting S1PR5 was purchased from Bioscien China; detailed sequences are provided in Supplementary Table 1. Cells were transfected using Lipofectamine 3000 according to the manufacturer's protocol, and knockdown efficiency was confirmed by Western blot analysis.

For microglia–NK cell co-culture experiments, siALDH3B1-transfected microglia were treated with various concentrations of 2-HD for 24 h, after which the conditioned medium was collected and applied to NK cells for downstream analyses. For NK–fibroblast or NK–endothelial cell co-culture assays, an 8.0-μm Transwell system (Corning) was used, with NK cells seeded in the upper chamber and fibroblasts or HUVECs in the lower chamber. After 24 h of co-culture, fibroblasts or endothelial cells were harvested from the lower chamber for subsequent analyses.

### Flow cytometry
Activated NK cells were incubated with 2-HD for one hour and then were washed with PBS twice. Washed cells were incubated with Fc receptor–blocking reagent (S0F0008-100T, STARTER) for 10 min at 4 °C and stained for 30 min at 4 °C in the dark, firstly with anti-S1PR5 antibody (ASR-015-50U, Alomone Labs, LTD) and were washed twice. Then the cells were stained with monoclonal antibodies against CD3 (FITC, 555339, BD Pharmingen), CD56 (BV421, 562751, BD Pharmingen), LIVE/DEAD viability dye (BV510, AB_2869572, BD Horizon™) and APC secondary antibody (A-10931, Invitrogen). Cells were washed, passed through a 40 μm strainer, and sorted on a BD FACSCelesta™.

### SPR assay
The binding of compounds to S1PR5 and their affinities were measured using SPR on a Biacore T200 instrument in collaboration with Shanghai InnoiHealth Biopharmaceutical Co.Ltd. Stock solutions of Siponimod and US-Louisville KY40299 were prepared in PBS-P containing 5% DMSO and diluted to the desired concentrations. Recombinant human S1PR5 protein (Q9H228, S1PR5_HUMAN Protein, Sphingosine

1-phosphate receptor 5) was reconstituted in ddH$_2$O and immobilized on an HC1500 sensor chip using standard amine-coupling chemistry. After protein immobilization, the chip surface was blocked and washed to remove unbound protein. Increasing concentrations of the test compounds were then flowed over the S1PR5-coated chip, with defined association and dissociation times, and binding responses were recorded for subsequent affinity analysis.

## Mutants generation

The *aldh3b1* knockout CRISPR line was generated as previously described[19]. Briefly, guide RNA (gRNA) targeting exon 2 of *aldh3b1* was designed with ZiFiT Targeter 4.2 and cloned into the pT7-gRNA vector (Addgene) for expression. (Supplementary Table S1). Cas9 mRNA was synthesized by transcribing a pT3TSnCas9n vector (Addgene) after linearizing with xbaI (Biolab). The T7 mMessage mMachine Kit (Invitrogen) and T3 MEGAshortscript (Invitrogen) were used to get gRNA and Cas9 mRNA, respectively, according to the suggestions of the manufacturer. At one cell stage, *Tg(fli1:EGFP)* embryos were injected with 200 pg/nl gRNA and 200 pg/nl Cas9-mRNA. Generated F0 fish were analyzed for germline transmission, crossed with *Tg(fli1: EGFP)* zebrafish and genotyped using Sanger sequencing and gel electrophoresis of PCR products. The genome and amino acid sequence were analyzed with benchling (benchiling.com).

The *s1pr5* morphants were established with morpholinos, which were designed and purchased from GENE TOOLS, LLC (Supplementary Table S1). All the morpholinos were diluted to 6 μg/ul with 0.1 mol KCl. One nanoliter of diluted morpholinos were injected into the yolk sack at the one-cell stage of the embryos. Their genotyping was performed by a Western blot experiment.

## Reverse-transcription quantitative polymerase chain Reaction analysis (RT-qPCR)

Total RNA from 20 larvae at different timepoints or one organ per fish were extracted with the RNeasy Mini Kit (Qiagen) according to the suggested protocols. 1 μg RNA was used to transcribe cDNA using the Maxima First Strand cDNA Synthesis Kit (Thermo Fisher Scientific), following the manufacturer's advice. Primers were designed with NCBI primer blast and produced by Sigma-Aldrich (Supplementary Table S1). RT-qPCR was done with Power SYBR Green PCR Master Mix Kit (Thermo Fisher Scientific) in 96 or 384 reaction plates and run by QuantStudio 3 or QuantStudio 5 Real-Time PCR System (Thermo Fisher Scientific).

## Western blot analysis

20 larvae/one adult organ were taken and homogenized in NP40 lysis buffer (150 mmol/L NaCl, 50 mmol/L, Tris-HCl pH 7.4, 1% NP40, 10 mmol/L EDTA, 10% glycerol, protease and phosphatase inhibitors) on ice for 30 min on a shaker. The lysate was diluted with 5 × Laemmli buffer and boiled at 95 °C for 5 min. Cellular proteins were resolved by SDS-PAGE and transferred to 0.2 μm nitrocellulose membrane (Amersham). The membranes were blocked with 5% BSA solution and incubated with primary antibodies (anti-Actin, Santa Cruz Biotechnology, sc-47778; anti-ALDH3B1, abcam, ab236673; anti-AMID, Santa Cruz Biotechnology, sc-377120; anti-GPX4, Santa Cruz Biotechnology, sc-166570; anti-S1PR5, Proteintech, 13874-1-AP), followed by secondary HRP-conjugated antibodies (rabbit anti-mouse, DAKO, P0260; Goat anti-Rabbit, Dako, P0448). Visualization by enhanced chemiluminescence (ECL) was were done by Western Lightning Plus ECL reagent (PerkinElmer) on a Vilber Fusion Solo S imaging system.

## Enzyme activity assay

At 5dpf, a batch of 120 larvae was anesthetized using a tricaine solution at a concentration of 0.003%, followed by rapid freezing in liquid nitrogen to prepare the samples. Samples were homogenized in ice-cold cytosolic lysis buffer (100 mM sodium phosphate, pH 7.4, 1 mM EDTA, 1 mM DTT and protease inhibitors) and centrifuged at 4 °C to obtain the supernatant. Protein concentration was determined using a BCA assay, and equal amounts of protein were used for all reactions. ALDH activity was measured spectrophotometrically by monitoring NAD(P)H production at 340 nm in a temperature-controlled microplate reader at 37 °C. Assays were performed in sodium-phosphate reaction buffer containing NAD(P)+ and saturating concentrations of reactive carbonyl substrates selected based on Michaelis–Menten kinetics. For the quantification of the maximum catalytic efficiency (V_max) of the ALDH, the required concentrations of various RCS were established, consistent with methodologies described in existing literature[68]. The specific reactive carbonyl species concentrations selected for analysis in the current study included: 5 mM for AA, 4 mM for 4HNE, 10 mM for 4-HHE[18], 10 mM for 2-HD, and 10 mM for tt-DDE[19].

## Microscopy and analysis of vascular alterations in larvae and adults

For the visualization of hyaloid vasculature, eyes were dissected and fixed in 4% PFA overnight at 5 dpf after larval zebrafish were anesthetized. Post-fixation, the eyes were washed in ddH$_2$O three times for 10 min each and then incubated in 0.3% Trypsin/EDTA solution (Gibco) buffered with TRIS HCl (1.5 M, pH 7.8) for 90 min at 37 °C[19]. After washing, hyaloids were dissected for imaging. Hyaloid vasculature images were captured by a confocal microscope (DM6000 B) with a Leica TCS SP5 DS scanner under a setting of 1024 × 1024 pixels, 0.6 μm Z-steps and 20 × 3 objective for statistical analysis.

The visualization and analysis of trunk vasculature were conducted as previously[19]. Briefly, larvae were incubated in E3 medium supplemented with 0.003% 1-phenyl-2-thiourea (PTU) (Sigma) to suppress pigmentation. At 4 dpf, larvae were put in 0.003% tricaine and individually placed in a 96-well plate for confocal microscopy examination of their trunk vasculature.

The methodology for analyzing the adult retina mirrors our previously established protocol[69]. The retina is carefully detached from the eye, placed on a slide, immersed in mounting medium, and covered with a cover slip for subsequent microscopic examination.

## Glucose measurement

For body glucose, a glucose assay kit (MAK263, Sigma-Aldrich) was employed according to the manufacturer's instructions. 20 larvae were collected at 5dpf as a sample and were homogenized in assay buffer with 20 G syringe. The fluorometric intensity was detected by a plate reader (Tecan Infinite M200).

To determine adult blood glucose levels, individual zebrafish were placed in separate boxes and transferred to an incubator for overnight fasting. The following day, blood glucose were collected from these fish, both in a fasted state and after being fed 0.5 g of flake food. The fish were euthanized prior to the collection of blood samples, which were then analyzed using a glucometer (Freestyle Abbott)[70].

## Pharmacological treatment of zebrafish embryos/larvae

Fertilized eggs were placed into 6-well plates, with each well containing 20 eggs and 5 mL of egg water. At 24 hpf, the embryos' chorions were carefully removed using tweezers, followed by immersion in various substrates and therapeutic agents for treatment and rescue experiments. The concentrations of the substances used included 0–200 μmol of 2-HD (FH23781, Biosynth), 10 mmol L-Carnosine (C9625, Sigma-Aldrich), 0–20 μmol of iFSP1 (SML2749, Sigma-Aldrich), 1 μmol Ferrostatin-1 (HY-100579, HY-100579), and 0–10 mmol A-971432 (5766, Bio-Techne/Tocris). Carnosine was dissolved directly in egg water, while all other compounds were dissolved in DMSO. The medium was replenished on a daily basis.

## RNA-Seq Analysis

For zebrafish samples, total RNA was extracted following the protocols outlined within the 'RT-QPCR' section of the manuscript. The

construction of the RNA libraries and the subsequent sequencing were carried out using the BGISEQ-500 platform, provided by the Beijing Genomic Institution (BGI), accessible online at www.bgi.com. Gene expression analysis was performed by the Core-Lab for microarray analysis, center for medical research (ZMF). Sequencing analyses were performed as described previously[18]. The RNA-Seq datasets produced in this study are available at GEO (Gene Expression Omnibus, NIH) under the accession number: (https://www.ncbi.nlm.nih.gov/geo/query/acc.cgi?acc=GSE264377).

Human RNA-Seq data were retrieved from the GEO database using the keyword 'human retina' and with a specific criterion that the selected probe sets must include the 'S1PR5' gene. Available datasets of retina disease characterized by neovascularization include AMD (GSE29801), DR (GSE29801, GSE102485 and GSE179568) and BRVO (GSE102485). To remove batch effects, the SVA (Surrogate Variable Analysis) R package was employed. After correcting for batch effects, the consolidated dataset was analyzed using R software (version 4.3, [https://www.r-project.org/]), following established RNA-Seq data analysis protocols. Epiretinal membrane is a pathologic fibrocellular tissue, so its samples were not considered as DR control and removed from GSE179568.

## Metabolomic analysis

Metabolomics detection and analysis was performed in cooperation with the Metabolomics Core Technology Platform (MCTP) of the Center for Organismal Studies (COS) of Heidelberg University via liquid chromatography-mass spectrometry (LC/MS) as published protocols[71]. One hundred zebrafish larvae or two eyes were collected as a sample.

Adenosine metabolites, free amino acids and thiols were extracted using 0.5 ml of 0.1 M HCl followed by sonication in an ice bath for 10 min. Homogenates were centrifuged twice at $16,400 \times g$ for 10 min at 4 °C to remove cellular debris, and the supernatants were used for subsequent metabolite analysis. Adenosine compounds were derivatized with chloroacetaldehyde and separated by reversed-phase chromatography using an Acquity BEH C18 column (150 mm × 2.1 mm, 1.7 μm; Waters) coupled to an Acquity H-class UPLC system (Waters). Metabolites were detected by fluorescence (excitation 280 nm, emission 410 nm). Free amino acids were quantified following derivatization with AccQ-Tag reagent (Waters) according to the manufacturer's instructions and detected by fluorescence (excitation 250 nm, emission 395 nm). Glutathione species were quantified after reduction with dithiothreitol and derivatization with monobromobimane, followed by UPLC–fluorescence detection (excitation 380 nm, emission 480 nm). Reduced glutathione levels were calculated by subtracting GSSG from total glutathione.

Data acquisition and processing were performed using Empower3 software (version 3.7.0, Waters). Quality control and multivariate statistical analyses, including PLS-DA and correlation mapping, were performed using R software.

## Lipid peroxidation assay

Lipid peroxidation was determined by lipid peroxidation (MDA) assay kit (MAK085, Sigma) using colorimetric tests according to the manufacturer's protocols. 20 larvae were collected as a sample with or without 2-HD addition.

## Iron assay kit

Iron quantification was performed using an iron assay kit (MAK025, Sigma), adhering to the provided instructions. 100 larvae as a sample were homogenized and incubated with an iron probe and reducer, enabling the measurement of $Fe^{2+}$ and total iron ($Fe^{2+} + Fe^{3+}$) concentrations.

## Molecular docking

The computational analysis of the binding affinities and interaction modes between 2-HD and S1PR5 was performed using the Schrödinger software suite, following the manufacturer's guidelines available at Schrödinger's website instructions (https://newsite.schrodinger.com/life-science/learn/white-papers/covdock/). Specifically, the Michael Addition reaction, predefined in Schrödinger's CovDock module, was utilized in our analysis without modification. The molecular structure of 2-HD was retrieved from the PubChem Compound database (CID5280541), while the three-dimensional (3D) coordinates for S1PR5, with a resolution of 2.2 Ångströms, were obtained from the Protein Data Bank (PDB ID 7YXA). For comparative purposes, positive control models, including ONO-5430608-S1PR5 and S1P-S1P3 complexes, were referenced from the Protein Data Bank using PDB ID 7YXA and PDB ID 7C4S, respectively. These models were analyzed in accordance with the methodology described in the published work[35].

## Enzyme-linked Immunosorbent Assay (ELISA) of S1P

The quantitative measurement of S1P was determined using a competitive inhibition enzyme immunoassay technique, employing an S1P ELISA kit (Catalog: MBS2700637, MyBiosource). Twenty zebrafish larvae or one eye per fish was considered as a sample and homogenized in fresh lysis buffer (catalog: MBS2090451, MyBiosource). The resulting supernatants were then incubated with a detection buffer and subjected to a series of washes as outlined in the assay's standard operating procedure provided by the kit manufacturer. Calculation of results was done through the establishment of a standard curve, which was plotted using the logarithm of known S1P concentrations.

## Single-cell RNA sequencing analysis

Single-cell gene expression datasets were obtained from studies by Collin et al. (GEO ID: GSE210543)[36] and Cowan et al.[37] (available at https://data.mendeley.com/datasets/sm67hr5bpm/1, accessed December 2023), including samples from adult, foveal, peripheral, and unaffected retinas.

Seurat R package (version 4.0.1, compatible with R version 4.1) was used to conduct the analysis[72]. Initial quality control measures included the exclusion of genes detected in fewer than three cells and cells exhibiting fewer than 200 genes with non-zero expression counts. Cells were further filtered to remove those with unique feature counts below 200 and those with mitochondrial gene counts exceeding 20% of their total gene counts. An upper limit for feature counts was established at 7000 for all datasets.

Integration of the datasets was achieved using the FindIntegrationAnchors method within the Seurat package, resulting in a combined dataset comprising 67,208 cells. Subsequent clustering utilized the UMAP (Uniform Manifold Approximation and Projection) technique. The classification of cell types within the clusters was based on established marker genes, with specific markers including GNGT2 and ARR3 for cone cells; NR2E3 and CNGA1 for rod cells; TRPM1, LRTM1, and PRKCA for bipolar cells; VAT1L and LNP1 for horizontal cells; NRXN2 and TFAP2A for amacrine cells; CD34 and CDH5 for vascular cells; FBLN1 and MGP for fibroblasts; MYL9, RGS5, and MYH11 for smooth muscle cells; C1QA and HLA-DPA1 for microglia; BEST1 and RPE65 for RPE cells; MLANA and MITF (excluding BEST1) for melanocytes; GFAP and RLBP1 for Müller/astrocytes; CD3 for T cells; and CD56 for NK cells.

## Statistical analysis

In this study, all the data were repeated more than three independent biological replicates. These data are presented as mean ± standard deviation (SD), as indicated in the figure legend. Statistical analysis of two groups was conducted using an unpaired Student's $t$ test, while one-way and two-way ANOVA were utilized for comparisons involving

multiple groups with one or two variables, respectively, and the log-rank test was used for survival rate analysis. Correlation matrices were generated using Pearson's or Spearman's correlation coefficients. Statistical analysis was performed by GraphPad Prism 8 and R software.

## Reporting summary

Further information on research design is available in the Nature Portfolio Reporting Summary linked to this article.

## Data availability

The RNA-Seq datasets produced in this study are available at GEO (Gene Expression Omnibus, NIH) under the accession number GSE264377. Other generated data supporting this paper are presented within the Supplementary Materials. Source data are provided in this paper.

## Code availability

The codes used for data analysis and figure generation in this study is available on GitHub at https://github.com/hasionwojoe/R-code and archived at Zenodo (https://doi.org/10.5281/zenodo.18772489).

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

## Acknowledgements

The study was supported by grants from Deutsche Forschungsgemeinschaft (CRC 1118 and project 517361638) and Shanghai Natural Foundation of Science (25ZR1402320). The authors thank Björn Hühn for zebrafish maintenance and Ina Schäfer from Live Cell Imaging Mannheim for excellent instruction of confocal microscope. The authors thank the Metabolomics Core Technology Platform of the Excellence

cluster "CellNetworks" (Heidelberg University), and the Deutsche Forschungsgemeinschaft (grant ZUK 40/2010–3009262) for support with UPLC-based metabolite quantification. The authors gratefully acknowledge the data storage service SDS@hd supported by the Ministry of Science, Research and the Arts Baden-Württemberg (MWK) and the German Research Foundation (DFG) through grant INST 35/1503-1 FUGG. We gratefully acknowledge the support of the Zebrafish Core Facility Mannheim.

## Author contributions

X.Q. designed this study, performed experiments, analyzed data and wrote the manuscript. R.G, Y.C, T.K, K.B., and X.Z. maintained the zebrafish line and performed experiments. B.L. generated *aldh3b1* knockout zebrafish. W.H. analyzed RNA-seq data. G.K. and G.P. performed metabolome studies, analyzed data and gave technological advice. J.M. and T.F. implemented and performed biochemical experiments. V.A. performed ScRNA-Seq analysis. I.H., J.S., and P.P.N. provided conceptual support. J.K. conceived and designed this study and wrote the manuscript. J.K. is the guarantor of this work, has full access to all data of the study and takes responsibility for the integrity and the accuracy of the data and the data analysis.

## Funding

## Competing interests

All the authors declare no competing interests.
