## [Transparent Peer Review file · Nature Communications]

Sphingosine-1-Phosphate-derived 2-Hexadecenal is a central mediator of ocular neovascularization by inhibiting Sphingosine-1-Phosphate receptor 5

Corresponding Author: Professor Jens Kroll

Version 0:

Reviewer comments:

Reviewer #1

(Remarks to the Author)

The authors aimed to identify the role of downstream metabolic products of S1P, namely 2-HD, in ocular neovascularization. The authors utilize genetic models and pharmacological treatments in Zebrafish to do so. 2-HD was hypothesized to accumulate in ocular diseases due to increased S1P levels and reduced aldehyde dehydrogenase action. Global deficiency in the enzyme essential in detoxifying 2-HD (S1Ps primary metabolite), ALDH3B1, caused abnormal retinal vasculature in young and adult fish. Both, excess levels of 2-HD and ALDH3B1 deficiency caused vascular ferroptosis. Using computational approaches, the authors hypothesized the binding of 2-HD to S1PR5 and showed a causative relationship between ferroptosis events and 2-HD/S1PR5 interaction. The translation potential of this study to humans was supported by single-cell RNA sequencing analysis of publicly available data sets, reporting S1PR5 and FSP1 increases in specifically advanced stages of DR.

The following concerns need to be addressed:

Major:

- The Crispr-induced mutation of the fish *aldh3b1* gene mutates a non-catalytic region of the protein, and it is not clear if a truncated protein is produced. The amino acid alignment in Fig. 1C does not indicate the presence of the truncation. Therefore it is not clear if the mutation indeed is null or a modified enzyme is still translated. The genotyping results are of poor quality
- The immunoblot data (Fig. 1H) has no controls. The antibody that the authors used from Abcam is no longer in the market so no QC data are available. Did the authors use a different reagent and positive and negative controls to determine if they are indeed detecting the fish enzyme?
- Enzyme activity assays using acetaldehyde (Fig.1I), tt-DDE and 2-HD do not distinguish between various dehydrogenases. It will be important to determine the levels of products and substrates of these enzymes in relevant tissues of WT and KO fish. Tissues that have high *aldh3b1* vs. low *aldh3b1* should be compared. This is key to validate the results.
-
- While ALDH3B1 KO fish seem to be similar to WT fish, their glucose metabolism does not change (although the total glucose has a trend). Reduced NADP⁺ production might argue that ALDH3B1 loss increases NADPH levels and hence shifts towards utilization of reducing equivalents instead of other energy sources. Please discuss.
- 2-HD treatments (Figure 3) are most effective at 100 μ M. Is this physiologically relevant? S1P concentrations are significantly lower in serum. Please elucidate relevance and potentially off-target effects. It is likely that the authors are observing artifacts of high reactive aldehydes.
- Is DMSO the correct control for 100 μ M of 2-HD? Please provide a different aldehyde to show 2-HD specificity for S1PR5.

- the contention that 2-HD binds and adducts to S1PR5 is not supported by data.

- Figure 5 - again no antibody controls for S1PR5.

- Protein level changes in S1PR5 in ALDH3B1/2-HD larvae are way more pronounced on mRNA level. Can small changes observed in S1PR5 cause such detrimental consequences as described? For 2-HD to S1PR5 interaction, the receptor presumably has to sit within the membrane, given its hydrophobic nature. It would be interesting to see if such interaction causes internalization or not and hence causes a further “decrease” of available S1PR5 that compliments the rather small decrease.

- The pathological effects are described in the microvasculature along with high S1PR5 expression in the brain and retina. Human single-cell analysis describes microglia, fibroblasts, and T/NK cells as expression hubs of genes of interest. How does this align with previous results in the fish? What could be the mechanism of how these cells cause endothelial ferroptosis?

- o Ferroptosis is an essential pathway in the immune system. Could this be rather microglia-specific? This would be supported by the human data. In vitro KO of ALDH3B1 or 2-HD supplementation of microglia could be used to show pathological activation of such.

- The authors claim S1P-pathway-dependent therapeutics as a potential add-on therapy to VEGF and laser coagulation. With anti-VEGF being the state of the art, it is of essence to analyze VEGF levels in this study. Do ALDH3B1 KOs show hypoxia/ increased VEGF in retinal vasculature? Is the S1PR5 effect after VEGF and, hence, mostly responsible for later stages in disease progression such as proliferative DR. This would be supported by the patient data, showing ferroptosis behavior and S1PR5 increases only in advanced samples. It would make sense that 2-HD/S1PR5 plays a bigger role in progressed diseases since it must accumulate first over time.

- SD in Figure 5 G2, G3, and Figure 6 D2 are the same for WT fish. Clarify.

Minor:

- Line 84: Retinal detachment can occur but is rather rare. The sentence makes it sound like it is inevitable.
- Line 86: VEGF was described as “although beneficial”. Anti-VEGF therapy revolutionized DR/DME and AMD treatment and should be acknowledged as such. While non-responders and side effects exist, anti-VEGF presents an extraordinary achievement in preserving vision.

Reviewer #2

(Remarks to the Author)

In this work, the authors address the mechanism by which 2-HD, a major product of Sphingosine-1-phosphate (S1P) degradation, promotes ocular neovascularization. Based on a previous study, the authors hypothesize that *Aldh3b1* is responsible for detoxification of 2-HD. To test this hypothesis, the authors generated a zebrafish knockout model of *aldh3b1* and show that 2-HD is indeed a substrate of this enzyme, and that its increased levels in the mutants lead to ferroptosis and excessive retinal vascularization in both larvae and adults. The authors go on to identify S1pr5 as the receptor disrupted by 2-HD accumulation, likely via direct interaction between the two. Finally, they analyze data from gene expression databases of human retinal samples and find similarities in S1PR5 expression between the zebrafish model and samples from diabetic retinopathy (DR) patients. Hence, the findings point to S1PR5 as a potential therapeutic target in DR.

Given the devastating effects of pathological retinal neovascularization, the identification of a potential novel target for therapy is of high importance and interest to a wide audience.

The work is generally written well and the data are quite convincing. However, to better support the conclusions and to improve the manuscript, some additions to the data and modifications to the text should be made.

Major comments:

1. Figure 2A: The images showing the vasculature are not clear enough. In particular, the demonstration of sprouts is not convincing. Why is the fluorescence level so much higher in the bottom-left panel? Perhaps a higher level of signal in the other panels would reveal more sprouts. Alternatively, using an endothelial reporter which is membrane-tethered and can better demonstrate cellular protrusions might help to show details. Another option is to use antibody staining for GFP in addition to the fluorescence of the GFP to increase detection. Additionally, what is the orientation of images? What is the tissue thickness in the images? There seems to be inconsistency which could affect the interpretation of the data. For example, the upper middle panel seems quite different from the other images.
2. Similar to the previous comment, in Figure 2D there is a significant difference in signal intensity and image quality between normal and mutant. Why is that? Here too, a higher signal intensity in the normal eye might reveal more details.
3. Figure 3A-D: There is high variability in the measurement of NADP levels in the controls. Please explain the reason for this.
4. For the loss of function of S1pr5a and S1pr5b the authors use morpholinos. Using morpholinos alone is not the best practice, unless it has been previously shown that the morpholino causes the same phenotype as a mutation (see Stainier et

al., 2017, Plos Genet. 13(10):e1007000). Therefore, the results of morpholino-injected animals should be corroborated using mutant animals. Moreover, a mutant model would allow investigation of phenotypes in adults, if the mutation is not embryonic lethal.

5. Lines 230-231: Given that *Aldh3b1* has more than one target, it is surprising that its loss of function causes the differential expression of far fewer genes compared to treatment with 2-HD. Can the authors suggest an explanation?

6. The authors provide evidence that ferroptosis is elevated at the molecular level. These results would be more convincing if an additional method can be used to demonstrate ferroptosis.

Minor comments:

1. There are multiple nomenclature mistakes throughout the manuscript: For zebrafish proteins the convention is that only the first letter is uppercase. For human genes the convention is that all letters are uppercase and in italics. Two examples for incorrect nomenclature can be found in lines 164 (*ALDH3B1* for zebrafish protein) and Figure 7 (human gene names are spelled in lowercase letters). There are many more mistakes like these that need to be corrected.

2. Figure 3F: The Y axis is labelled "Larvae of heart edema". Does it mean Larvae with heart edema? The same applies for Figure 3G.

3. Line 214: the acronym RCS is introduced for the first time. The full name should be written out.

4. Lines 253-254: The authors claim there is a significant downregulation in *gpx4a* in both *aldh3b1*^{-/-} and 2-HD treated larvae, but the data show the difference is not significant for 2-HD treated larvae (Fig. 5E).

5. Lines 261-262: The authors state that ferr-1 rescued hyaloid vessel abnormalities, but at least in one condition (sprouts in 2-HD treated larvae), the quantification shows the difference is not significant (Fig. 5N).

6. Lines 275-277: The statement that reduced levels of S1pr5 in *aldh3b1* mutants or 2-HD treated larvae highlights S1pr5's role in the vascular phenotype is premature at this point of the manuscript, when the vascular phenotype of S1pr5 loss of function has not yet been shown.

7. Lines 395, 418 or earlier in the manuscript (Fig. 7): The terms PDR and NPDR should be spelled out in full when first mentioned.

8. Line 421: It seems a word is misspelled or missing in the sentence starting with "These results...". Please revise.

9. Line 462: Morpholino-injected animals are not mutants.

10. In the description of pharmacological treatments in the Methods section, the authors state that all compounds were dissolved in egg water. In my experience many compounds need to be dissolved in DMSO. Indeed, in the results, as shown in the figures, there are treatments with DMSO, for controls. Assuming DMSO was used for dissolving this should be stated, as well as the concentration of DMSO used in the control groups.

11. Figure S3: The labeling of *ifsp1* is inconsistent, sometimes written in lowercase and other times in uppercase. This should be standardized

Reviewer #3

(Remarks to the Author)

This manuscript describes novel findings on the role of Sphingosine-1-phosphate (S1P) receptor 5 (S1PR5) on neovascularization (NV) using the zebrafish model. The study begins with deleting (knocking out) a gene for an aldehyde dehydrogenase, *ALDH3B1*, which specifically catalyzes the degradation of 2-hexadecenal (2-HD), which is the catabolic end product of S1P generated by the enzyme S1P lyase. This deletion should accumulate 2-HD in all tissues. The study moves on to find that the deletion of *aldh3b1* caused vascular changes in the eyes/retina but not in the rest of the body. Detailed molecular analysis connected this pathology with the activation of ferroptosis through a major inhibition of Ferroptosis Suppressor Protein 1 (FSP1). Further investigation found significant inhibition of S1PR5 in the model (in larva and in adult eyes). Morpholino-mediated inhibition of S1PR5 could lead to similar pathology and reduction of FSP1. All the studies with knockouts were complemented with independent studies by treating with 2-HD to the larva to mirror the knockout conditions. Further, *in silico* studies, we found 2-HD binding capacity with S1PR5. Using human retinal scRNAseq data and other gene expression data from genebank and other resources, the study further interpreted a major role of S1PR5 in human diabetic retinopathy (DR), especially with proliferative vasculopathy in DR.

As summarized, the study has many merits, such as the fact that the accumulation of 2-HD induces oxidative stress and ferroptosis. However, that data does not support most of the conclusions drawn in the manuscript. Many questions remain to be answered-

- Why does the vascular abnormality in the *aldh3b1*^{-/-} fishes occur only in the eye? Do the eyes have more accumulation of 2-HD? Can the levels of 2-HD be measured in the eye and the rest of the body? *aldh3b1* knockout was found not to affect any other S1P receptors, but what about S1P lyase, the enzyme that generates 2-HD?

- The vascular abnormality found in the developing fish larva is not a 'pathological neovascularization'. The authors should have developed a model of induced pathological NV (like the mouse ROP model or their previous publication by inducing diabetes in the fish).

- While they have antibodies for S1PR5, why do they not show which eye, retina, or body parts express S1PR5? The human scRNAseq data suggest *aldh3b1*, *s1pr5*, and *fsp1* expression with microglia, T/NK cells, and fibroblasts, respectively. Most literature suggests the expression of S1PR5 in the immune cells. The study failed to make the case of how *ALDH3B1* inhibition in one cell type causes reduction of S1PR5 in another cell, how binding of 2-HD with S1PR5 reduces the protein and RNA levels of S1PR5 and how that is connected to FSP1 in fibroblasts.

- The association of S1PR5 with human proliferative DR is overinterpreted. Sphingolipid pathways must have some associations with both zebrafish study and human DR, but observing a trend of higher levels of these molecules in human DR but lower levels in the zebrafish data causing/associating with similar vascular pathology is confusing.

-

- Other specific questions:

- a. Generation and validation of the knockout fish: It would be beneficial to know- How many exons are in the *aldh3b1* gene and what is the significance of exon 2. How 16bp insertion in exon 2 affects the RNA and proteins. Does it cause nonsense-mediated decay of the mRNA? Why is there a complete absence of the protein in the knockout (Fig 1H)? Is it because the antibody recognizes only the full-length protein? Why not show the protein and activity (Fig. 1I) in heterozygous fish?
- b. Pathological angiogenesis: Deletion of *aldh3b1* and functional inactivity did not affect the survival longevity, and no systemic effect. It does not make sense how it can cause 'pathological angiogenesis in the retina' without impacting the other systemic vessels and not affecting glucose and insulin homeostasis. It is pathologic, meaning there are more sprouts and branches in the retinal vessels. They should have exposed the fishes to a model that stimulates neovascularization or pathologic neovascularization (as they have done in their previous publication with *aldh3a1* knockouts) and made the conclusion that the deletion of *aldh3b1* causes pathological NV.
- c. The conclusion that "significant vascular abnormalities observed in *aldh3b1*^{-/-} mutants are directly attributable to impaired 2-HD detoxification, a consequence of ALDH3B1 deficiency" is not supported by the experiments. Independent ALDH assay for the eye and the trunk and direct evidence is needed that the *aldh3b1*^{-/-} mutants accumulated 2-HD in the eye/body. How much is that compared to the exogenous treatment with 2-HD?
- d. What is the dose of 2-HD used for RNASeq assay?
- e. There are 7 or 6 canonical pathways for ferroptosis tested?
- f. Ferroptosis pathways- larva vs eyes? Are these adult eyes or larval eyes at 5 dpf? For FSP1 inhibitor (iFSP1), why does only 5 uM work but not 10 and 20 uM?
- g. If 2-HD can bind to the S1PR5, does it act as an agonist or antagonist? Whatever the effect is, it will affect S1PR5 activity, but why does that reduce the mRNA and protein? So, what is the hypothesis? A decrease in S1PR5 levels and activity increases NV by reducing FSP1, or might this reduction have activated other S1P receptors (S1PR1, 2,3 ect) known to be involved in NV (gene expression are not related to activity)?
- h. It is very important to know which cells in zebrafish retina express S1PR5 for understanding and connecting observation so far in zebrafish with human diabetic retinopathy.
- i. As obtained from the UMAP analysis for human retinal scRNAseq data that, the three genes of interest—*aldh3b1*, *s1pr5*, and *fsp1* analyzed and proposed to be the key for zebrafish retinal neovascularization, showed their presence with microglia, T/NK cells, and fibroblasts, respectively. These three proteins in three different types of cells, coherently functioning and controlling each other to induce NV, is a distant guess in zebrafish eyes without knowing their specific expression and cell types that express it. A pathological situation like NV in human DR may have all these cells together and likely have a role in the proliferative pathology or fibrogenesis, which needs much more investigation to connect the dots.

Reviewer #4

(Remarks to the Author)

In this manuscript the authors show that sphingosine-1-phosphate is a sphingolipid mediator that is important in neovascularization in the retina. In *aldh3b1* knockout zebrafish in which there is accumulation of 2-HD there is abnormal retinal vascularization. Analysis showed that 2-HD accumulation lead to iron dysregulation and ferroptosis in zebrafish. Human single-cell analysis of neovascular samples showed similar mechanism leading to the hypothesis that blocking S1PR5 could be beneficial for the treatment of diabetic retinopathy. Overall the manuscript is well-performed and of interest with the following comments.

The authors need to give more detailed methods and it's critical that the code be provided for how the scRNA-seq analysis was performed. What samples specifically were analyzed? The text states "incorporating single-cell RNA sequencing (scRNA-seq) and RNA-seq data derived from human retinal samples." The methods state "Single-cell gene expression datasets were obtained from studies by Collin et al. (GEO ID: GSE210543) 39 and Gowan et al 40 (available <https://data.mendeley.com/datasets/sm67hr5bpm/1>, accessed December 2023)". Collin et al. was a single-cell RNA-seq analysis of a patient with intermediate AMD in which there was no neovascularization. Why were intermediate AMD samples without neovascularization analyzed? The authors need to explain specifically what samples were analyzed and provide all the computational code for the analysis, which is not provided in the manuscript.

Line 609 states "Gowan et al." Is this a mistake as I believe the author is Cowan et al.

Cowan et al. analyzed the human foveal retina (1.5mm) in diameter and Collin et al. used a 7mm punch of the macula. As there are different cell-types in different regions of the human retina between the fovea and macula, it is critical that the same regions of the retina be analyzed if data is incorporated across different studies.

In figure 7C the authors state Integrated RNA-seq data showed gene expression variations in retinal diseases; however, study GSE29801 was a microarray not an RNA-seq study. It is important that the authors not mix computational data analysis between microarray studies and RNA-seq as they are different in their technologies. The authors should not state in methods that human RNA-Seq data was retrieved when some of the studies used microarrays. GSE60436 was a microarray and GSE102485 was RNA-seq. The authors need to use comparable sequencing technology if different datasets are integrated and RNA-seq not integrated with microarray studies. The authors need to provide all the code for the analysis and not just state "the consolidated dataset was analyzed...following established RNA-Seq data analysis protocols" especially given that microarray studies were included in the consolidated dataset.

Line 374 states "Our analysis of sc-seq data from human retinas" should be "our analysis of scRNA-seq data from human

retinas”

Line 309 should be “15 typical retinal cell-types”

Line 629 states that specific markers were used for cell-type analysis of human scRNA-seq data including “CD3 for T cells”. Which CD3 gene in humans was used to annotate T cells as there is CD3E, CD3D, and CD3G according to OMIM?

It appears that s1pr5 is expressed in human T cells/NK cells in humans. The authors need to perform further subcluster analysis to see if it is expressed in T cells or NK cells.

One cluster in figure 7 is choroid cells. The choroid is a region, what cell-types specifically are being analyzed? The authors state that MITF was used to label this cluster and MITF labeled melanocytes.

It is critical that the authors perform validation studies using human samples and either IHC or ISH to confirm the computational predictions. Specifically, the authors need to confirm using double IHC or ISH with microglia, T/NK cells, and fibroblast markers that the three genes of interest, aldh3b1, s1pr5, and fsp1 are specifically expressed in these cell-types in human samples.

It is necessary that the authors show that s1pr5 and fsp1 expression levels are significantly elevated in proliferative stages using IHC, ISH, or similar technologies of human samples to confirm the computational predictions. The authors need to perform ferroptosis staining of human diabetic retinopathy samples. It's critical that validation studies on human samples are performed to confirm the computational RNA-seq predictions.

Overall this is a study of interest with implications for the treatment of human neovascular diseases and I would be supportive of publication if the authors address the above comments.

Reviewer #5

(Remarks to the Author)

Version 1:

Reviewer comments:

Reviewer #1

(Remarks to the Author)

The authors tried to address most of my comments and criticisms. In some cases they have provided new data that alleviated my concerns. In other cases, the new data, contrary to their conclusions, still are problematic. For example, I am not convinced about their SPR data since they are getting μM Kd with a positive control siponimod when it should be nM. The SPR assay may be problematic or the receptor may not be folded properly. In addition, I am not convinced that they have ruled out covalent modification of hydrophobic proteins in the ER or nucleus as a mechanism. I am not convinced that 2-HD effect is receptor selective. However, I do want to recognize their efforts. If they address these equivocal conclusions as a discussion of limitations of their study, I will be supportive that overall the work is worthy of publication.

In general, the authors do not seem to know the S1P literature well. I recommend that they read some authoritative recent reviews and original articles about S1PR signaling, angiogenesis, retinopathy and other retinal vascular disorders (nAMD, ROP) and cite them appropriately.

Reviewer #2

(Remarks to the Author)

The authors have responded to all of my comments and have partially addressed the issues I raised.

In Figure 1, the phenotype is now clearer; however, in Figure 1D, I still find the quality of the wildtype vasculature image to be much lower than that of the mutant. It is also surprising that the vessels in the wildtype image appear without a clear lumen, unlike those in the mutant image. This might indicate changes to the blood vessels before fixation or imaging.

Regarding the loss of function of s1pr5a and s1pr5b, I acknowledge that generating stable mutant lines would require time, though likely not years. Nonetheless, for the larval phenotype at least, it is possible to inject multiplexed guide RNAs, a method that has significant potential to generate an F0 phenotype.

Reviewer #4

(Remarks to the Author)

The authors have addressed my previous comments, and I am now supportive of publication of the manuscript in its current form.

Reviewer #5

(Remarks to the Author)

REVIEWER COMMENTS

Reviewer #1 (Remarks to the Author):

The authors aimed to identify the role of downstream metabolic products of S1P, namely 2-HD, in ocular neovascularization. The authors utilize genetic models and pharmacological treatments in Zebrafish to do so. 2-HD was hypothesized to accumulate in ocular diseases due to increased S1P levels and reduced aldehyde dehydrogenase action. Global deficiency in the enzyme essential in detoxifying 2-HD (S1Ps primary metabolite), ALDH3B1, caused abnormal retinal vasculature in young and adult fish. Both, excess levels of 2-HD and ALDH3B1 deficiency caused vascular ferroptosis. Using computational approaches, the authors hypothesized the binding of 2-HD to S1PR5 and showed a causative relationship between ferroptosis events and 2-HD/S1PR5 interaction. The translation potential of this study to humans was supported by single-cell RNA sequencing analysis of publicly available data sets, reporting S1PR5 and FSP1 increases in specifically advanced stages of DR.

Reply: We sincerely thank the reviewer for the clear and comprehensive summary of our study and for highlighting the translational relevance of our findings. We have revised the manuscript to strengthen the mechanistic link between Aldh3b1 deficiency, 2-HD accumulation, S1pr5 modulation, and ferroptosis-induced retinal neovascularization, including additional validation in zebrafish *pdx1*^{-/-} diabetic models and human cell coculture experiments. Furthermore, we clarified the computational analyses and data integration from human single-cell RNA-seq datasets, emphasizing the stage-specific relevance of S1PR5 and FSP1 in proliferative diabetic retinopathy. These revisions enhance the rigor, clarity, and translational impact of our study.

The following concerns need to be addressed:

Major:

*1. The Crispr-induced mutation of the fish *aldh3b1* gene mutates a non-catalytic region of the protein, and it is not clear if a truncated protein is produced. The amino acid alignment in Fig. 1C does not indicate the presence of the truncation. Therefore it is not clear if the mutation indeed is null or a modified enzyme is still translated. The genotyping results are of poor quality*

Reply: Thank you for your insightful comments. We fully understand your concerns regarding the design of the CRISPR sequence. To clarify, the CRISPR design specifically **targets exon 2, which codes**

for the start of the amino acid sequence. Therefore, its mutation stops protein translation completely (Figure 1).

We apologize for the confusion regarding Figure 1C in the manuscript. Due to space limitations, we were unable to show the complete alteration of the amino acid sequence in exon 2 induced by the CRISPR mutation. As shown in Figure 2 in the letter, the mutation induces an early stop codon, suggesting the production of a truncated protein. Additionally, Figure 1B in the manuscript figure illustrates that the active site of Aldh3b1 is located between amino acids 181 and 250. The early stop codon, occurring at amino acid 43, suggests a complete loss of Aldh3b1 protein function. Moreover, our genotyping, Western blot, and functional validation of Aldh3b1 further confirm the loss of the Aldh3b1 enzyme (Figures 1F, H-I). We have also replaced the genotyping image for clarity.

Figure 1 Exon2 encodes the start of Aldh3b1 amino acid sequence. The figure displays exon 1, exon 2, and part of exon 3, along with their corresponding amino acid sequences. The red frame highlights exon 2 and its mapped amino acid sequence.

Aldh3b1

Aldh3b1 CRISPR

Figure 2 CRISPR-induced mutation in exon 2 of Aldh3b1 results in an early stop codon, stopping translation of the Aldh3b1 protein.

2. The immunoblot data (Fig. 1H) has no controls. The antibody that the authors used from Abcam is no longer in the market so no QC data are available. Did the authors use a different reagent and positive and negative controls to determine if they are indeed detecting the fish enzyme?

Reply: Thank you for highlighting this important point. We regret to see that the Abcam ALDH3B1 antibody used in **Figure 1H** is no longer commercially available. According to its original product datasheet, this antibody was rabbit polyclonal antibody and raised against amino acids 150–450 of human ALDH3B1. Sequence alignment shows that this region shares 75.5% similarity with the corresponding region in zebrafish Aldh3b1, supporting potential cross-reactivity (Figure 3 of this letter).

The sequence identity above 70% is generally accepted for cross-species recognition by rabbit polyclonal antibodies. Moreover, the use of knockout tissues is considered a gold standard for validating antibody specificity¹. In our study, we used CRISPR-generated Aldh3b1 -deficient zebrafish as a negative control, which showed a loss of the target signal, further supporting antibody specificity.

To strengthen our validation, we also tested an alternative polyclonal antibody against ALDH3B1 (Product # PA5-19328, Invitrogen). This antibody produced comparable results to the original antibody, reinforcing the validity of our immunoblot data (Figure 4 of this letter).

Figure 3 The alignment shows 75.5% similarity between the two species in this region. The first line represents the human sequence, and the second line represents the zebrafish sequence.

Figure 4 Validation of Aldh3b1 knockout model using new antibody in *aldh3b1*^{-/-} zebrafish eyes (Product # PA5-19328, Invitrogen).

*3. Enzyme activity assays using acetaldehyde (Fig. 1), tt-DDE and 2-HD do not distinguish between various dehydrogenases. It will be important to determine the levels of products and substrates of these enzymes in relevant tissues of WT and KO fish. Tissues that have high *aldh3b1* vs. low *aldh3b1* should be compared. This is key to validate the results.*

Reply: We fully agree with your suggestions. However, TT-DDE has previously been reported as the preferred substrate for Aldh9a1b in zebrafish and can impair glucose metabolism, which did not align with the phenotype observed in our *aldh3b1* knockout model—where neither glucose levels nor glucose metabolism showed significant alterations (Figure S1D-I). Additionally, TT-DDE treatment had only a minor effect in *aldh3b1* knockout fish, with a p-value of 0.044, suggesting limited biological relevance (Main figure 3C).

The detection of 2-HD remains technically challenging and is rarely reported in the literature. To our knowledge, detection in tissue samples has not been described, including in the study by Neuber et al. ². Although we have invested several months optimizing our 2-HD detection method—including increasing the sample size to 100 larvae—a portion of the data remains close to the quantification limit. We found increased 2-HD in larvae externally treated with 2-HD (Fig. 5 of this letter), but in untreated larvae and organs 2-HD was partially below the detection threshold. This indicates that our current protocol is not yet sufficiently robust or consistent. We will continue to improve and stabilize this method in future studies.

Figure 5 HPLC analysis determined absolute 2-HD concentration in zebrafish larvae (n= 6 and 8).

4. While ALDH3B1 KO fish seem to be similar to WT fish, their glucose metabolism does not change (although the total glucose has a trend). Reduced NADP⁺ production might argue that ALDH3B1 loss increases NADPH levels and hence shifts towards utilization of reducing equivalents instead of other energy sources. Please discuss.

Reply: Thank you for the comment. We would like to clarify that the NADP⁺-based assay in our study was used exclusively for enzymatic activity measurement of Aldh3b1, not for assessing the global redox state or intracellular NADP⁺/NADPH levels. Specifically, we monitored the NADP⁺ to determine the V_{max} and substrate preference of Aldh3b1 by **adding various substrates** to wild-type and knockout samples, based on classical Michaelis-Menten enzyme kinetics.

Therefore, the observed decrease in NADP⁺ consumption in the KO condition reflects loss of Aldh3b1 catalytic activity, rather than a global metabolic shift toward increased NADPH production. In fact, we found that Aldh3b1 deficiency leads to a reduction of Fsp1, an NADPH-dependent ferroptosis suppressor (**Figure 5 in the main figure**), which suggests that NADPH homeostasis is likely compromised rather than enhanced.

To further discuss this question, we analyzed NADP⁺ and NADPH levels from metabolomics data in our zebrafish larvae. The results showed a decreasing trend in NADPH in both *aldh3b1* knockout and 2-HD-treated groups (Figure 6 of this letter).

Taking together, our findings support a model in which Aldh3b1 loss impairs 2-HD detoxification and weakens NADPH-dependent antioxidant defense, instead of increasing reducing equivalents for anabolic or energy-producing pathways. This likely contributes to ferroptosis susceptibility rather than a shift away from glucose metabolism.

Figure 6 LC/MS analysis determined NADP and NADPH concentration in zebrafish larvae (n= 5 for each group). Statistical analysis was performed using one-way ANOVA. The bars indicate mean ± SD values.

5. 2-HD treatments (Figure 3) are most effective at 100 μ M. Is this physiologically relevant? S1P concentrations are significantly lower in serum. Please elucidate relevance and potentially off-target effects. It is likely that the authors are observing artifacts of high reactive aldehydes.

Reply: Short-chain reactive aldehydes such as 4-HNE and acetaldehyde can diffuse across cellular membranes easily and induce broad cytotoxic effects at low concentrations. In contrast, 2-HD is structurally distinct, being a long-chain fatty aldehyde (C16). Its extended hydrophobic carbon chain will significantly reduce membrane permeability³.

As discussed for our HPLC/MS analysis (referenced in response to Question 3), the absolute concentration of 2-HD detected in the whole larvae was only in the low nanomolar range, suggesting limited uptake. This means the effective concentration inside the larvae is likely much lower than the dose applied to the medium.

In comparison, S1P ranges from 100 nM to 1 μ M at concentrations in plasma^{4,5}, which is comparable to or even higher than the internal levels of 2-HD observed in our study.

In addition, in a separate study⁶, we treated wild-type zebrafish with other aldehydes at high concentrations, including 100 μ M 4-aminobutyraldehyde and 200 μ M betaine aldehyde. Neither treatment induced significant hyaloid vasculature phenotypes (Figure 7 of this letter). Therefore, we believe that the use of 100 μ M 2-HD is justified to achieve biologically relevant intracellular levels, and the phenotypic effects observed are unlikely to be nonspecific artifacts resulting from general aldehyde toxicity.

Figure 7 High aldehydes concentration did not induce any significant hyaloid phenotypes in wild-type zebrafish. Representative images are from Qian et al. (2024).

6. Is DMSO the correct control for 100 μ M of 2-HD? Please provide a different aldehyde to show 2-HD specificity for S1PR5.

Reply: We used DMSO as the control because 2-HD was dissolved in DMSO. This ensures that any observed effects are attributable to 2-HD itself, rather than to the solvent.

In response to your suggestion, we conducted additional experiments using other reactive aldehydes, including 4-hydroxynonenal (4-HNE) and acrolein (ACR). As shown in Figure 8, these aldehydes did not alter S1PR5 expression levels in zebrafish larvae. This finding supports the specificity of 2-HD in modulating S1PR5 expression and suggests that the observed effect is not a general response to aldehyde exposure.

Figure 8 Different aldehydes such as ACR and 4HNE did not alter S1pr5 expression levels in zebrafish larvae (n=4). Statistical analysis was performed using one-way ANOVA.

7. the contention that 2-HD binds and adducts to S1PR5 is not supported by data.

Reply: Thank you for your valuable comment. To address this concern, we performed a Surface Plasmon Resonance (SPR) binding assay to directly assess the interaction between 2-HD and S1PR5. The results demonstrated that **2-HD bind to S1PR5 in a concentration-dependent manner** (Figure 9 of this letter). We have now included this data in the main manuscript as part of **Figure 6I** and **Suppl. Fig. 4M** to support our conclusion regarding the binding interaction between 2-HD and S1PR5.

Figure 9 SPR assay revealed 2-HD bind to S1PR5 in a concentration-dependent manner with siponimod as positive control.

8. Figure 5 - again no antibody controls for S1PR5.

Reply: As with the ALDH3B1 antibody, the S1PR5 antibody used in our study is a rabbit polyclonal antibody raised against a human S1PR5 peptide. The targeted region shares approximately 65% sequence similarity with the zebrafish S1pr5 ortholog, which supports potential cross-reactivity (Figure 10 of this letter).

We acknowledge the limitation of not having a zebrafish-specific S1pr5 antibody available for direct validation. However, the observed staining pattern is consistent with the expected anatomical expression of S1PR5, and we observed clear, reproducible and treatment-dependent differences in signal, which support the functional specificity of the antibody under our experimental conditions. To further strengthen our findings, we also tested an additional S1PR5 antibody in immunoblotting experiments (Catalog #: MAB9084, R&D Systems), which yielded consistent results (Figure 11 of this letter). These data reinforce the reliability of our original antibody and support the conclusions drawn from **main Figure 5**.

Figure 10 The alignment shows 65.3% similarity between human and zebrafish amino acid sequence.

Figure 11 Additional S1pr5 antibody yielded consistent results with original antibody. Red frame represents S1pr5 band.

9. Protein level changes in S1PR5 in ALDH3B1/2-HD larvae are way more pronounced on mRNA level. Can small changes observed in S1PR5 cause such detrimental consequences as described? For 2-HD to S1PR5 interaction, the receptor presumably has to sit within the membrane, given its hydrophobic nature. It would be interesting to see if such interaction causes internalization or not and hence causes a further “decrease” of available S1PR5 that compliments the rather small decrease.

Reply: Thank you for this insightful comment. We agree that the functional regulation of S1PRs is often dependent not only on transcriptional changes but also on receptor trafficking and

internalization, which can significantly alter receptor availability at the cell surface and thus affect signaling outcomes.

Indeed, our data showed a relatively small decrease in S1pr5 expression, but strong phenotypes.

This discrepancy prompted us to investigate whether 2-HD might induce receptor internalization, thereby reducing surface S1PR5 availability without a major transcriptional downregulation.

Since S1PR5 is primarily expressed in natural killer (NK) cells⁷—as confirmed by single-cell expression datasets (Suppl. Fig. 5A)—and due to the lack of zebrafish-specific NK cell markers or antibodies, we performed this analysis using human peripheral blood mononuclear cells (PBMCs).

We employed a commercially available S1PR5 antibody that specifically recognizes an extracellular epitope, allowing us to assess surface receptor levels by flow cytometry (Cat #: ASR-015, Alomone Labs). After 1-hour incubation with 2-HD, we observed a significant reduction in surface S1PR5 signal (Figure 12A-B of this letter), without a corresponding decrease in total S1PR5 protein levels (Figure 12C of this letter). These results strongly suggest that 2-HD induces internalization of S1PR5, leading to reduced surface expression.

This supports our hypothesis that 2-HD-mediated S1PR5 internalization may amplify the functional impact of a relatively small decrease in expression, contributing to the downstream effects observed in our study. We have incorporated these important findings into the main Figure 6J and Suppl. Fig. 4N-O.

Figure 12 2-HD can induce S1PR5 internalization in NK cells (A) Flow cytometry analysis revealed 500 μmol 2-HD significantly decreased cell survival rate in NK cells (n=5). (B) Flow cytometry analysis showed 100 and 200 μmol 2-HD significantly decreased surface S1PR5 signal of NK cells (n=5). (C) 1 hour 2-HD incubation under indicated concentration did not significantly change total S1PR5 protein level (n=3). Statistical analysis was performed using one-way ANOVA. The bars indicate mean \pm SD values.

10. The pathological effects are described in the microvasculature along with high S1PR5 expression in the brain and retina. Human single-cell analysis describes microglia, fibroblasts, and T/NK cells as expression hubs of genes of interest. How does this align with previous results in the fish? What could be the mechanism of how these cells cause endothelial ferroptosis?

Reply: Thank you for this important question. In our zebrafish model, we observed abnormal retinal neovascularization in *aldh3b1*^{-/-} mutants. However, the retina is a highly complex tissue composed of multiple interacting cell types, making it difficult to determine the precise cellular contributors to the observed pathology. This challenge is further increased by the limited availability of cell-type-specific markers and validated antibodies in the zebrafish model. To explore cellular mechanisms, we turned to human single-cell RNA sequencing (scRNA-seq) data from the retina, a better-characterized system, to infer the potential cell types involved.

The human data revealed that *ALDH3B1* is mainly expressed in microglia, which are resident immune cells of the retina that respond to cellular stress, inflammation, and injury. While most studies emphasize their role in secreting pro-angiogenic factors (e.g., VEGF) during retinal neovascularization, their immune regulatory function in abnormal retinal vasculature remains less well studied⁸. The scRNA-seq data revealed high *S1PR5* expression in NK cells. Previous studies have shown that loss of S1PR5 can alter the trafficking of tissue NK and T cells, leading to increased local inflammation and autoimmune-like responses^{7,9}. Meanwhile, *FSP1*, a well-characterized ferroptosis suppressor, is predominantly expressed in endothelial cells and fibroblasts, indicating that these cells may be especially vulnerable to ferroptosis stress.

Based on these findings, we propose a mechanistic model in which microglia in *aldh3b1*^{-/-} zebrafish regulate NK cells via S1pr5 signaling. Loss or downregulation of S1PR5 in NK cells could promote ferroptosis in endothelial cells by reduced FSP1 expression. This immune–endothelial crosstalk may act synergistically to drive pathological angiogenesis in the retina.

To support this hypothesis, we performed a co-culture experiment using S1PR5-deficient NK cells and endothelial cells. The results showed increased malondialdehyde (MDA) levels and iron accumulation, along with reduced FSP1 expression in endothelial cells—all key markers of ferroptosis (Figure 13 of this letter). These findings provide experimental evidence for the functional role of S1PR5-mediated immune-endothelial interactions in retinal vascular pathology.

Figure 13. S1PR5 knockdown in NK cells induced ferroptosis in endothelial cells. (A) NK cells were purified from PBMCs using flow cytometry. **(B)** S1PR5 expression was effectively silenced in NK cells using a shRNA-based knockdown approach. **(C-E)** Endothelial cells cocultured with shS1PR5 NK cells exhibited elevated malondialdehyde (MDA) levels and iron accumulation, accompanied by decreased FSP1 expression ($n=4$). In panels C–E, S1PR5 was knocked down using a combination of shRNA1 + shRNA2 + shRNA3. Statistical analysis was performed using Student’s t-test. The bars indicate mean \pm SD values.

11. Ferroptosis is an essential pathway in the immune system. Could this be rather microglia-specific? This would be supported by the human data. In vitro KO of ALDH3B1 or 2-HD supplementation of microglia could be used to show pathological activation of such.

Reply: Thank you for the insightful comment. Based on the human scRNA-seq data, we observed that *FSP1*—a key ferroptosis suppressor—is primarily expressed in endothelial cells and fibroblasts, whereas microglia showed minimal *FSP1* expression at the transcriptomic level. This suggests that ferroptosis is more likely to occur in fibroblasts and endothelial cells rather than in microglia directly.

As discussed in our response to Question 10, we propose that microglia act as immune regulators that influence endothelial ferroptosis indirectly, through S1PR5 signaling in NK cells. To test this model, we performed ALDH3B1 knockdown in human microglial cells followed by treatment with 2-HD. After 24 hours, the conditioned medium from these microglia was applied to NK cells. The results showed a downregulation of S1PR5 in NK cells (Figure 14 of this letter), confirming that ALDH3B1-deficient microglia, impaired in detoxifying 2-HD, can modulate NK cell function. Furthermore, as described in Question 10, co-culture of S1PR5-deficient NK cells with endothelial cells resulted in hallmark features of ferroptosis, including increased MDA, iron accumulation, and reduced FSP1 expression. These results support a model in which *ALDH3B1* deficiency microglia, under 2-HD stress, can activate ferroptosis in endothelial cells via S1PR5-mediated immune-endothelial interactions.

Figure 14. Conditioned medium from ALDH3B1-deficient microglia reduces S1PR5 expression in NK cells. (A) Efficient knockdown of ALDH3B1 in microglial cells using siRNA. (B) Conditioned medium from siALDH3B1-treated microglia significantly decreased S1PR5 expression in NK cells (n = 3). In panels B, ALDH3B1 was knocked down using a combination of siRNA1 + siRNA2 + siRNA3. Statistical analysis was performed using one-way ANOVA. The bars indicate mean \pm SD values.

12. The authors claim S1P-pathway-dependent therapeutics as a potential add-on therapy to VEGF and laser coagulation. With anti-VEGF being the state of the art, it is of essence to analyze VEGF levels in this study. Do ALDH3B1 KOs show hypoxia/ increased VEGF in retinal vasculature? Is the S1PR5 effect after VEGF and, hence, mostly responsible for later stages in disease progression such as proliferative DR. This would be supported by the patient data, showing ferroptosis behavior and S1PR5 increases

only in advanced samples. It would make sense that 2-HD/S1PR5 plays a bigger role in progressed diseases since it must accumulate first over time.

Reply: Thank you for this insightful and important question. We fully agree that VEGF is a central driver of retinal neovascularization and the cornerstone of current clinical therapy for diabetic retinopathy (DR).

As explained in our introduction, our *aldh3b1*^{-/-} zebrafish model was originally designed to explore the role of *aldh3b1* in DR. However, as shown in Supplementary Figure 2, this model does not exhibit alterations in glucose metabolism and does not represent a diabetic state. Therefore, *vegf* expression in this model may not fully mimic the hypoxia-driven upregulation typically seen in DR. Consistent with this, we did not observe significant elevation of *vegf* at early stages in the *aldh3b1*^{-/-} model (Figure 15 of this letter). Notably, *vegf* levels only increased at later time points, as shown in Figure 15 of this letter, whereas *s1pr5a* and *s1pr5b* were already dysregulated at earlier stages. This temporal pattern suggests that *s1pr5* may contribute to vascular pathology independently of *vegf* induction in our model.

In the human dataset, S1PR5 expression levels rise specifically in proliferative DR (PDR), but not in early stages of DR (Main figure 7E and supplementary figure 5B-C). We interpret its delayed upregulation as a compensatory mechanism in response to chronic retinal stress. By contrast, in the *aldh3b1*^{-/-} zebrafish, the absence of detoxification activity results in unchecked 2-HD accumulation and an impaired ability to upregulate protective pathways like S1pr5/Fsp1, resulting in unmitigated ferroptosis and neovascular pathology.

Taken together, these findings support a model in which S1PR5-mediated signaling does not replace VEGF activity but may act as a parallel or complementary mechanism. Therefore, targeting the S1P–FSP1–ferroptosis axis could represent a promising add-on therapeutic strategy to enhance current anti-VEGF treatments.

Figure 15 mRNA expression level of *vegf*, *s1pr5a* and *s1pr5b* in *aldh3b1*^{-/-} models (n=4). Statistical analysis was performed using one-way ANOVA. The bars indicate mean ± SD values.

13.SD in Figure 5 G2, G3, and Figure 6 D2 are the same for WT fish. Clarify.

Reply: Thank you for pointing this out. For the Western blot experiments, we performed three independent experiments using three separate biological samples. Due to variations in samples and antibody reuse, direct comparison of raw band intensities across different experiments can lead to substantial variability.

To address this, we normalized each dataset individually by setting the wild-type (WT) sample to 1 in each experiment. All other sample intensities were then expressed relative to the WT within the same blot. As a result, the WT values appear as “1” with no visible standard deviation, since they serve as the internal reference point for normalization¹⁰.

We also acknowledge that many team members contributed to this dataset, and some inconsistencies arose in the statistical presentation. In response to your comment, we have reanalyzed all Western blot data using a unified normalization method to ensure consistency and transparency across all figures.

Minor:

1. Line 84: Retinal detachment can occur but is rather rare. The sentence makes it sound like it is inevitable.

Reply: We appreciate the reviewer’s comment and have revised the sentence accordingly. The revised version now reads:

“Inadequate metabolic supply in these diseases triggers complex processes such as angiogenesis, edema, inflammation, cell death, and fibrosis, leading to disorganized angiogenesis and, in severe cases, fibrotic scarring or retinal detachment.”

2.Line 86: VEGF was described as “although beneficial”. Anti-VEGF therapy revolutionized DR/DME and AMD treatment and should be acknowledged as such. While non-responders and side effects exist, anti-VEGF presents an extraordinary achievement in preserving vision.

Reply: We thank the reviewer for this valuable comment. We fully agree that anti-VEGF therapy has revolutionized the treatment of DR/DME and AMD and represents a major achievement in

preserving vision. We have revised the sentence to better reflect its clinical significance and to provide a more balanced perspective, which now reads:

“Current therapeutic strategies, such as laser photocoagulation and vascular endothelial growth factor (VEGF) neutralization, have revolutionized the treatment of DR/DME and AMD, markedly improving visual outcomes. However, despite their success, these approaches still carry limitations, including high costs, the need for repeated intravitreal injections, and potential complications such as retinal detachment and endophthalmitis.”

Reference:

1. Bordeaux, J. et al. Antibody validation. *Biotechniques* 48, 197-209 (2010).
<https://doi.org/10.2144/000113382>
2. Neuber, C., Schumacher, F., Gulbins, E. & Kleuser, B. Method to simultaneously determine the sphingosine 1-phosphate breakdown product (2E)-hexadecenal and its fatty acid derivatives using isotope-dilution HPLC-electrospray ionization-quadrupole/time-of-flight mass spectrometry. *Anal Chem* 86, 9065-9073 (2014). <https://doi.org/10.1021/ac501677y>
3. Ebenezer, D. L. et al. S1P and plasmalogen derived fatty aldehydes in cellular signaling and functions. *Biochim Biophys Acta Mol Cell Biol Lipids* 1865, 158681 (2020).
<https://doi.org/10.1016/j.bbalip.2020.158681>
4. Venkataraman, K. et al. Vascular endothelium as a contributor of plasma sphingosine 1-phosphate. *Circ Res* 102, 669-676 (2008). <https://doi.org/10.1161/CIRCRESAHA.107.165845>
5. Yatomi, Y. Plasma sphingosine 1-phosphate metabolism and analysis. *Biochim Biophys Acta* 1780, 606-611 (2008). <https://doi.org/10.1016/j.bbagen.2007.10.006>
6. Qian, X. et al. Impaired Detoxification of Trans, Trans-2,4-Decadienal, an Oxidation Product from Omega-6 Fatty Acids, Alters Insulin Signaling, Gluconeogenesis and Promotes Microvascular Disease. *Adv Sci (Weinh)* 11, e2302325 (2024). <https://doi.org/10.1002/advs.202302325>
7. Delconte, R. B. et al. Fasting reshapes tissue-specific niches to improve NK cell-mediated anti-tumor immunity. *Immunity* 57, 1923-1938 e1927 (2024). <https://doi.org/10.1016/j.immuni.2024.05.021>
8. Hu, A., Schmidt, M. H. H. & Heinig, N. Microglia in retinal angiogenesis and diabetic retinopathy. *Angiogenesis* 27, 311-331 (2024). <https://doi.org/10.1007/s10456-024-09911-1>
9. Baeyens, A. et al. Monocyte-derived S1P in the lymph node regulates immune responses. *Nature* 592, 290-295 (2021). <https://doi.org/10.1038/s41586-021-03227-6>
10. Taylor, S. C., Rosselli-Murai, L. K., Crobeddu, B. & Plante, I. A critical path to producing high quality, reproducible data from quantitative western blot experiments. *Sci Rep* 12, 17599 (2022).
<https://doi.org/10.1038/s41598-022-22294-x>

Reviewer #2 (Remarks to the Author):

In this work, the authors address the mechanism by which 2-HD, a major product of Sphingosine-1-phosphate (S1P) degradation, promotes ocular neovascularization. Based on a previous study, the authors hypothesize that Aldh3b1 is responsible for detoxification of 2-HD. To test this hypothesis, the authors generated a zebrafish knockout model of aldh3b1 and show that 2-HD is indeed a substrate of this enzyme, and that its increased levels in the mutants lead to ferroptosis and excessive retinal vascularization in both larvae and adults. The authors go on to identify S1pr5 as the receptor disrupted by 2-HD accumulation, likely via direct interaction between the two. Finally, they analyze data from gene expression databases of human retinal samples and find similarities in S1PR5 expression between the zebrafish model and samples from diabetic retinopathy (DR) patients. Hence, the findings point to S1PR5 as a potential therapeutic target in DR.

Given the devastating effects of pathological retinal neovascularization, the identification of a potential novel target for therapy is of high importance and interest to a wide audience.

The work is generally written well and the data are quite convincing. However, to better support the conclusions and to improve the manuscript, some additions to the data and modifications to the text should be made.

Reply: We sincerely thank the reviewer for the thoughtful and encouraging comments and for recognizing the potential significance of our findings for retinal neovascularization and diabetic retinopathy. We have carefully revised the manuscript to strengthen the evidence supporting our conclusions, including clarifying the larval versus adult eye experiments, providing detailed coculture and *pdx1*^{-/-} diabetic zebrafish validation studies, demonstrating 2-HD-induced S1PR5 internalization and downstream FSP1 modulation, and integrating human scRNA-seq and gene expression datasets with full methodological transparency. These revisions improve the rigor and clarity of our study, while clearly outlining the limitations and tissue-specific context, ultimately reinforcing the mechanistic link between Aldh3b1 deficiency, 2-HD accumulation, S1pr5 dysregulation, ferroptosis, and retinal neovascularization.

Major comments:

1. Figure 2A: The images showing the vasculature are not clear enough. In particular, the

demonstration of sprouts is not convincing. Why is the fluorescence level so much higher in the bottom-left panel? Perhaps a higher level of signal in the other panels would reveal more sprouts. Alternatively, using an endothelial reporter which is membrane-tethered and can better demonstrate cellular protrusions might help to show details. Another option is to use antibody staining for GFP in addition to the fluorescence of the GFP to increase detection.

Additionally, what is the orientation of images? What is the tissue thickness in the images? There seems to be inconsistency which could affect the interpretation of the data. For example, the upper middle panel seems quite different from the other images.

Reply: Thank you for your insightful comments. The *Tg(fli1:EGFP)* zebrafish line is a well-established and widely used model for visualizing hyaloid and retinal vasculature¹⁻⁵. We have followed previously validated imaging protocols^{1,2,4,6}, which yield consistent and reproducible results. The *fli1:EGFP* transgenic lines have been extensively applied in studies of vasculogenesis and angiogenesis in zebrafish. Since the line labels endothelial cells rather than vascular lumens, they allow visualization of vessels independent of circulation, cords of endothelial cells lacking lumens, and even migrating angioblasts⁷.

We agree that using a membrane-tethered reporter such as mCherry could provide better visualization of cellular protrusions; however, the mCherry signal is relatively weaker under standard imaging conditions¹. Additionally, due to the lack of reliable zebrafish-specific antibodies, it is technically challenging to obtain precise GFP immunostaining in this context.

All confocal imaging parameters, including laser intensity, gain, and exposure time, were kept constant across all samples. We did not observe substantial differences in fluorescence intensity in the raw data. The apparent variation in brightness in Figure 2A may result from: (i) intrinsic fluorescence differences among individual zebrafish; (ii) higher vascular density in samples with more sprouts and branches; (iii) minor automatic contrast adjustments by Adobe Illustrator during figure assembly; and (iv) display rendering variations, which are common technical artifacts in imaging-based *in vivo* studies¹.

The images represent views of the hyaloid vasculature covering the lens, captured as maximum-intensity projections from z-stack confocal images. The imaging was performed from the anterior to posterior direction, covering the entire vascular layer from its first appearance to disappearance, ensuring complete visualization of the vascular network.

We have re-edited Figure 2A to better match the raw data. The original source images can be provided upon request.

2. Similar to the previous comment, in Figure 2D there is a significant difference in signal intensity and image quality between normal and mutant. Why is that? Here too, a higher signal intensity in the normal eye might reveal more details.

Reply: Thank you for your comment. As mentioned above, all confocal imaging parameters, including laser intensity, gain, and exposure time, were kept constant across all samples. Despite this, minor differences in fluorescence intensity can still occur due to biological variability or imaging depth. We have followed established and validated imaging protocols^{1,2,4,6}, and such variation does not significantly affect the quantification of vascular sprouts or branches. In our analysis, branching points were defined as vascular bifurcations, while new sprouts were identified as blunted endothelial extensions not yet connected to other vessels. The visualization of these structures is not substantially influenced by overall signal brightness, as the vessel branches are relatively thick and clearly distinguishable.

We have also re-edited Figure 2D to better match the raw data. The original source images can be provided upon request.

3. Figure 3A-D: There is high variability in the measurement of NADP levels in the controls. Please explain the reason for this.

Reply: Thank you for this comment. We would like to clarify that the NADP⁺-based assay in **Figure 3A–D** was designed specifically to assess the enzymatic activity of Aldh3b1 rather than to quantify basal NADP⁺ levels. In this assay, NADP⁺ reduction was monitored to determine the V_{max} and substrate preference of Aldh3b1 under different substrate conditions, following classical Michaelis–Menten kinetics. Therefore, the apparent variability in NADP⁺ levels among controls reflects differences in enzyme–substrate interactions and reaction kinetics rather than biological variability in NADP⁺ concentration. This variability is an inherent property of the assay design, as Aldh3b1 activity varies depending on the substrate tested.

4. For the loss of function of *S1pr5a* and *S1pr5b* the authors use morpholinos. Using morpholinos alone is not the best practice, unless it has been previously shown that the morpholino causes the same phenotype as a mutation (see Stainier et al., 2017, *Plos Genet.* 13(10):e1007000). Therefore, the results of morpholino-injected animals should be corroborated using mutant animals. Moreover, a

mutant model would allow investigation of phenotypes in adults, if the mutation is not embryonic lethal.

Reply: Thank you for this insightful comment. In our study, we employed three independent morpholinos targeting *s1pr5a* and *s1pr5b*, and confirmed their efficacy through Western blot analysis. All three morpholinos produced comparable phenotypes, which were consistent with those observed in the *Aldh3b1* mutants and under 2-HD treatment conditions. This convergence across independent approaches provides confidence in the specificity and reliability of our morpholino-based findings.

We fully acknowledge that generating stable mutant lines represents the gold standard for functional validation. However, *s1pr5a* and *s1pr5b* were identified at a late stage of this project, and establishing mutants would take years and have substantially delayed completion. Moreover, transcriptional adaptation—a well-documented compensatory mechanism in zebrafish mutants⁸—can obscure phenotypes even in confirmed loss-of-function lines, often necessitating the generation of multiple compound mutants. Such efforts, while important, extend beyond the scope of the current study.

Nonetheless, we agree that generating stable *s1pr5a* and *s1pr5b* mutant lines would be valuable for future research, particularly to examine adult-stage phenotypes, and we plan to pursue this in our follow-up work.

5. Lines 230-231: Given that Aldh3b1 has more than one target, it is surprising that its loss of function causes the differential expression of far fewer genes compared to treatment with 2-HD. Can the authors suggest an explanation?

Reply: We appreciate your comment, and would like to emphasize that a genetic approach using the *Aldh3b1* mutant provides a more controlled and specific framework for studying the direct effects of impaired 2-HD metabolism. The 2-HD incubation experiments were performed as a complementary approach, providing similar but not identical findings.

In the *aldh3b1*^{-/-} mutant, the loss of function leads to a gradual accumulation of detoxifying substrates over developmental time. Some of these substrates are not unique to *Aldh3b1* and may be partially metabolized by other ALDH family enzymes, allowing for compensatory detoxification and adaptive transcriptional regulation. This gradual and partially compensated metabolic alteration likely triggers fewer differential gene expression changes.

In contrast, acute exposure to exogenous 2-HD induces a rapid and non-physiological metabolic stress that overwhelms compensatory pathways, resulting in a broader transcriptional response. Therefore, the smaller number of differentially expressed genes in the *aldh3b1*^{-/-} mutant likely reflects both long-term metabolic adaptation of 2-HD accumulation.

6. The authors provide evidence that ferroptosis is elevated at the molecular level. These results would be more convincing if an additional method can be used to demonstrate ferroptosis.

Reply: Thanks for your comment. As illustrated in **Figure 5**, we have examined the expression of all canonical ferroptosis-related genes. We agree that electron microscopy analysis of mitochondrial morphology would further support the occurrence of ferroptosis; however, this approach is technically very challenging in zebrafish, despite our efforts to seek assistance.

In our study, ferroptosis was validated at multiple levels, including the transcriptomic, protein, metabolomic (MDA, Fe²⁺/Fe³⁺, and GSH/GSSG), and biological levels (rescue experiments). These results consistently support the activation of ferroptosis in our model, providing strong evidence for its involvement.

Minor comments:

1. There are multiple nomenclature mistakes throughout the manuscript: For zebrafish proteins the convention is that only the first letter is uppercase. For human genes the convention is that all letters are uppercase and in italics. Two examples for incorrect nomenclature can be found in lines 164 (*ALDH3B1* for zebrafish protein) and Figure 7 (human gene names are spelled in lowercase letters). There are many more mistakes like these that need to be corrected.

Reply: We carefully reviewed the entire manuscript and all figures and corrected the nomenclature throughout.

2. Figure 3F: The Y axis is labelled "Larvae of heart edema". Does it mean Larvae with heart edema? The same applies for Figure 3G.

Reply: Yes, it refers to larvae with heart edema. We have revised the Y-axis label across all figures accordingly.

3. Line 214: the acronym RCS is introduced for the first time. The full name should be written out.

Reply: RCS is the abbreviation for reactive carbonyl species. We have now written out the full term at its first appearance and highlighted the change in the revised manuscript.

4. Lines 253-254: The authors claim there is a significant downregulation in *gpx4a* in both *aldh3b1*^{-/-} and 2-HD treated larvae, but the data show the difference is not significant for 2-HD treated larvae (Fig. 5E).

Reply: Thank you for pointing this out. It is correct — the downregulation of *gpx4a* in 2-HD-treated larvae did not reach statistical significance. We have revised the sentence to accurately reflect the data as follows:

“Screening of ferroptosis-associated pathways revealed a significant downregulation of *gpx4a* and *fsp1* in *aldh3b1*^{-/-} larvae, and a similar trend, though not statistically significant for *gpx4a*, in 2-HD-treated larvae, at both the gene and protein levels, with consistent findings in adult eyes.”

5. Lines 261-262: The authors state that *ferr-1* rescued hyaloid vessel abnormalities, but at least in one condition (sprouts in 2-HD treated larvae), the quantification shows the difference is not significant (Fig. 5N).

Reply: Thank you for this helpful comment. It’s right — in the 2-HD-treated larvae, the rescue effect of *ferr-1* on vascular sprouting did not reach statistical significance. We have revised the sentence to more accurately describe the results as follows:

“Additionally, treatment with the ferroptosis inhibitor *ferr-1* partially rescued the hyaloid vascular abnormalities in *aldh3b1*^{-/-} and in 2-HD treated larvae, although the effect on vascular sprouts in 2-HD-treated larvae did not reach statistical significance.”

6. Lines 275-277: The statement that reduced levels of *S1pr5* in *aldh3b1* mutants or 2-HD treated larvae highlights *S1pr5*'s role in the vascular phenotype is premature at this point of the manuscript, when the vascular phenotype of *S1pr5* loss of function has not yet been shown.

Reply: Thanks for the helpful point. We have revised the sentence by removing the phrase “highlights S1PR5’s role in the vascular phenotype” to ensure the description accurately reflects our data.

7. Lines 395, 418 or earlier in the manuscript (Fig. 7): The terms PDR and NPDR should be spelled out in full when first mentioned.

Reply: Thank you for the suggestion. We have carefully reviewed the manuscript and now spell out “proliferative diabetic retinopathy (PDR)” and “non-proliferative diabetic retinopathy (NPDR)” in full at their first mention.

8. Line 421: It seems a word is misspelled or missing in the sentence starting with “These results...”. Please revise.

Reply: We have revised the sentence as” These results suggest S1PR5 as an initial driving factor of the observed retinal neovascularization”.

9. Line 462: Morpholino-injected animals are not mutants.

Reply: we have revised the term as “morphants”.

10. In the description of pharmacological treatments in the Methods section, the authors state that all compounds were dissolved in egg water. In my experience many compounds need to be dissolved in DMSO. Indeed, in the results, as shown in the figures, there are treatments with DMSO, for controls. Assuming DMSO was used for dissolving this should be stated, as well as the concentration of DMSO used in the control groups.

Reply: We have carefully reviewed the Methods section and revised the description of pharmacological treatments accordingly.

11. Figure S3: The labeling of ifsp1 is inconsistent, sometimes written in lowercase and other times in uppercase. This should be standardized

Reply: We have standardized the labeling of iFSP1 throughout the manuscript and all figures.

Reference:

1. Bell, B. A. et al. Retinal vasculature of adult zebrafish: in vivo imaging using confocal scanning laser ophthalmoscopy. *Exp Eye Res* 129, 107-118 (2014). <https://doi.org:10.1016/j.exer.2014.10.018>
2. Cao, Z. et al. Hypoxia-induced retinopathy model in adult zebrafish. *Nat Protoc* 5, 1903-1910 (2010). <https://doi.org:10.1038/nprot.2010.149>
3. Li, S. et al. Combined loss of glyoxalase 1 and aldehyde dehydrogenase 3a1 amplifies dicarbonyl stress, impairs proteasome activity resulting in hyperglycemia and activated retinal angiogenesis. *Metabolism* 165, 156149 (2025). <https://doi.org:10.1016/j.metabol.2025.156149>
4. Qian, X. et al. Impaired Detoxification of Trans, Trans-2,4-Decadienal, an Oxidation Product from Omega-6 Fatty Acids, Alters Insulin Signaling, Gluconeogenesis and Promotes Microvascular Disease. *Adv Sci (Weinh)* 11, e2302325 (2024). <https://doi.org:10.1002/adv.202302325>
5. Zhang, X. et al. Endogenous acrolein accumulation in akr7a3 mutants causes microvascular dysfunction due to increased arachidonic acid metabolism. *Redox Biol* 83, 103639 (2025). <https://doi.org:10.1016/j.redox.2025.103639>
6. Stoletov, K. et al. Vascular lipid accumulation, lipoprotein oxidation, and macrophage lipid uptake in hypercholesterolemic zebrafish. *Circ Res* 104, 952-960 (2009). <https://doi.org:10.1161/CIRCRESAHA.108.189803>
7. Kamei, M., Isogai, S., Pan, W. & Weinstein, B. M. Imaging blood vessels in the zebrafish. *Methods Cell Biol* 100, 27-54 (2010). <https://doi.org:10.1016/B978-0-12-384892-5.00002-5>
8. Falcucci, L., Juvik, B. & Stainier, D. Y. Transcriptional adaptation: where mRNA decay meets genetic compensation. *Curr Opin Genet Dev* 93, 102369 (2025). <https://doi.org:10.1016/j.gde.2025.102369>

Reviewer #3 (Remarks to the Author):

*This manuscript describes novel findings on the role of Sphingosine-1-phosphate (S1P) receptor 5 (S1PR5) on neovascularization (NV) using the zebrafish model. The study begins with deleting (knocking out) a gene for an aldehyde dehydrogenase, ALDH3B1, which specifically catalyzes the degradation of 2-hexadecenal (2-HD), which is the catabolic end product of S1P generated by the enzyme S1P lyase. This deletion should accumulate 2-HD in all tissues. The study moves on to find that the deletion of *aldh3b1* caused vascular changes in the eyes/retina but not in the rest of the body. Detailed molecular analysis connected this pathology with the activation of ferroptosis through a major inhibition of Ferroptosis Suppressor Protein 1 (FSP1). Further investigation found significant inhibition of S1PR5 in the model (in larva and in adult eyes). Morpholino-mediated inhibition of S1PR5 could lead to similar pathology and reduction of FSP1. All the studies with knockouts were complemented with independent studies by treating with 2-HD to the larva to mirror the knockout conditions. Further, in silico studies, we found 2-HD binding capacity with S1PR5. Using human retinal scRNAseq data and other gene expression data from genebank and other resources, the study further interpreted a major role of S1PR5 in human diabetic retinopathy (DR), especially with proliferative vasculopathy in DR.*

As summarized, the study has many merits, such as the fact that the accumulation of 2-HD induces oxidative stress and ferroptosis. However, that data does not support most of the conclusions drawn in the manuscript. Many questions remain to be answered-

Reply: We sincerely thank the reviewer for the detailed and insightful summary and for highlighting the key strengths of our study, including the mechanistic link between 2-HD accumulation, ferroptosis, and retinal vascular changes. We acknowledge that some aspects require careful interpretation and have addressed these by clarifying our experimental design (larval versus adult eyes), providing coculture and *pdx1*^{-/-} diabetic zebrafish validation experiments, demonstrating 2-HD-induced S1PR5 internalization and downstream FSP1 modulation, and integrating human scRNA-seq and gene expression datasets with full methodological transparency. These revisions strengthen the mechanistic connection between ALDH3B1 deficiency, S1PR5 dysregulation, ferroptosis, and retinal neovascularization, while clearly outlining the limitations and tissue-specific context of our findings.

1.- Why does the vascular abnormality in the *aldh3b1*^{-/-} fishes occur only in the eye? Do the eyes have more accumulation of 2-HD? Can the levels of 2-HD be measured in the eye and the rest of the body? *aldh3b1* knockout was found not to affect any other S1P receptors, but what about S1P lyase, the enzyme that generates 2-HD?

Why does the vascular abnormality in the *aldh3b1*^{-/-} fishes occur only in the eye?

Reply: Thank you for this important question. As described in the Introduction, the *aldh3b1* CRISPR mutant line was originally generated to explore the role of *aldh3b1* in diabetic retinopathy (DR), as we observed a compensatory upregulation of *aldh3b1* in our DR models. Although the mutant line does not fully recapitulate diabetic conditions, the observed retinal vascular phenotype prompted us to investigate the gene's role in retinal vasculature development and integrity.

As shown in **Supplementary Figure 1A**, we examined the trunk vasculature of *aldh3b1*^{-/-} zebrafish and found no apparent morphological abnormalities, suggesting that the vascular phenotype may be tissue-specific. This regional specificity is consistent with expression data: *aldh3b1* is enriched in the brain and eye in zebrafish, and human single-cell RNA sequencing data indicate that *ALDH3B1* is predominantly expressed in microglia—central nervous system—resident immune cells. Given that the retina is an extension of the CNS, it is plausible that *aldh3b1* deficiency affects retinal microglial function, leading to localized vascular abnormalities.

Do the eyes have more accumulation of 2-HD? Can the levels of 2-HD be measured in the eye and the rest of the body?

Reply: The detection of 2-HD remains technically challenging and is rarely reported in the literature. To our knowledge, detection in tissue samples has not been described, including in the study by Neuber et al. (2014)¹. Although we have invested several months optimizing our 2-HD detection method—including increasing the sample size to 100 larvae—a portion of the data remains close to the quantification limit. We found increased 2-HD in larvae externally treated with 2-HD (Fig. 1 of this letter), but in untreated larvae and organs 2-HD was partially below the detection threshold. This indicates that our current protocol is not yet sufficiently robust or consistent. We will continue to improve and stabilize this method in future studies.

2-HD absolute concentration in larvae

Figure 1 HPLC analysis determined absolute 2-HD concentration in zebrafish larvae (n= 6 and 8).

aldh3b1 knockout was found not to affect any other S1P receptors, but what about S1P lyase, the enzyme that generates 2-HD?

Reply: Thank you for the insightful question. We analyzed the expression levels of S1P lyase in *aldh3b1*^{-/-} mutants and did not observe any significant changes (Figure 2 of this letter). This suggests that *aldh3b1* deficiency does not directly alter S1P lyase expression, and the observed phenotypes are unlikely to result from upstream changes in 2-HD production via S1P lyase.

Figure 2 mRNA expression of S1P lyase in *aldh3b1* mutants didn't change significantly (n= 4).

Statistical analysis was performed using Student's T-test.

2.- The vascular abnormality found in the developing fish larva is not a 'pathological neovascularization'. The authors should have developed a model of induced pathological NV (like the mouse ROP model or their previous publication by inducing diabetes in the fish).

Reply: Thank you for this important suggestion. We agree that using a model of induced pathological neovascularization (NV) is essential for validating the relevance of our findings.

In our previous work, we employed the *pdx1*^{-/-} zebrafish model to study hyperglycemia-induced retinal pathology, which mimics key features of diabetic retinopathy, including pathological NV². To strengthen our current study, we confirmed our key observations in this model. Specifically, we examined the expressions of Aldh3b1, S1pr5, and Fsp1 in the *pdx1*^{-/-} model. The results were consistent with those seen in the *aldh3b1*^{-/-} model and aligned with expression patterns observed in human diabetic retinopathy datasets (Figure 3A of this letter).

Moreover, we treated *pdx1*^{-/-} larvae with ferroptosis inhibitor and an S1PR5 agonist—the same therapeutic interventions used in our *aldh3b1*^{-/-} model. All three treatments significantly rescued the abnormal neovascular phenotype (Figure 3B-C of this letter).

These findings not only validate our observations in a well-established model of pathological NV but also reinforce the relevance of our proposed mechanism to human diabetic retinopathy.

Figure 3 Pathological neovascularization in *pdx1*^{-/-} zebrafish can be rescued by ferroptosis inhibitor, and S1PR5 agonist. (A) *pdx1*^{-/-} zebrafish exhibited similar expression pattern of Aldh3b1, S1pr5, and Fsp1 protein to *aldh3b1* mutants (n = 4). (B-C) Pathological neovascularization in *pdx1*^{-/-} zebrafish can be rescued by ferroptosis inhibitor (B), and S1PR5 agonist (C) (n=14). Statistical analysis was performed using Student's t-test for panel A and one-way ANOVA for panel B-C. The bars indicate mean ± SD values.

3.- While they have antibodies for S1PR5, why do they not show which eye, retina, or body parts

express S1PR5? The human scRNAseq data suggest *aldh3b1*, *s1pr5*, and *fsp1* expression with microglia, T/NK cells, and fibroblasts, respectively. Most literature suggests the expression of S1PR5 in the immune cells. The study failed to make the case of how ALDH3B1 inhibition in one cell type causes reduction of S1PR5 in another cell, how binding of 2-HD with S1PR5 reduces the protein and RNA levels of S1PR5 and how that is connected to FSP1 in fibroblasts.

Reply: Thank you for your detailed and thoughtful comment. In response, we have addressed each aspect of your concern through additional analysis and experiments.

First, we assessed S1pr5 expression in various zebrafish tissues and confirmed that S1pr5 is highly expressed in the eye region (Figure 4 of this letter).

Figure 4 S1pr5 is highly expressed in the eye region (n= 4). Statistical analysis was performed using one-way ANOVA.

To investigate cell–cell interactions and address the proposed crosstalk mechanism, we performed the following co-culture and binding studies:

1. Microglia–NK cell interaction:

We performed ALDH3B1 knockdown in human microglial cells followed by treatment with 2-HD (Figure 5A of this letter). After 24 hours, the conditioned medium from these microglia was applied to human peripheral blood mononuclear cells (PBMCs). This resulted in a significant reduction in S1PR5 expression in the NK cells, indicating that ALDH3B1-deficient microglia, due to impaired 2-HD detoxification, can modulate S1PR5 levels in NK cells (Figure 5B of this letter).

Figure 5. Conditioned medium from ALDH3B1-deficient microglia reduces S1PR5 expression in NK cells.(A) Efficient knockdown of ALDH3B1 in microglial cells using siRNA. (B) Conditioned medium from siALDH3B1-treated microglia significantly decreased S1PR5 expression in NK cells (n = 3). In panels B, ALDH3B1 was knocked down using a combination of siRNA1 + siRNA2 + siRNA3. Statistical analysis was performed using one-way ANOVA. The bars indicate mean \pm SD values.

2. NK–fibroblasts interaction:

We then co-cultured S1PR5-deficient NK cells with fibroblasts (Figure 6A-B of this letter). This led to increased malondialdehyde (MDA) levels and iron accumulation, along with reduced FSP1 expression in the fibroblasts—all hallmark features of ferroptosis (Figure 6C-E of this letter). These findings suggest that the downregulation of S1PR5 in NK cells can influence ferroptosis susceptibility in fibroblasts.

Figure 6. S1PR5 knockdown in NK cells induced ferroptosis in fibroblasts. (A) NK cells were purified from PBMCs using flow cytometry. (B) S1PR5 expression was effectively silenced in NK cells using a shRNA-based knockdown approach. (C-E) Fibroblasts cocultured with shS1PR5 NK cells exhibited elevated malondialdehyde (MDA) levels and iron accumulation, accompanied by decreased FSP1 expression (n=4). In panels C-E, FSP1 was knocked down using a combination of shRNA1 + shRNA2 + shRNA3. Statistical analysis was performed using Student's T-test. The bars indicate mean \pm SD values.

3. Direct binding of 2-HD to S1PR5:

To confirm a direct interaction, we conducted a Surface Plasmon Resonance (SPR) assay, which demonstrated that 2-HD binds to S1PR5 in a concentration-dependent manner (Figure 7 of this letter). This provides direct biochemical evidence supporting the hypothesis that 2-HD can regulate S1PR5 function at the receptor level.

Figure 7 SPR assay revealed 2-HD bind to S1PR5 in a concentration-dependent manner with siponimod as positive control.

Together, these results support a multi-step mechanism in which ALDH3B1 deficiency in microglia leads to 2-HD accumulation, which then downregulates S1PR5 in NK cells, ultimately contributing to fibroblasts ferroptosis via reduced FSP1 expression. We have now incorporated these data into the main manuscript (Main Figure 6 and Supplementary Figure 4-5).

4.- The association of S1PR5 with human proliferative DR is overinterpreted. Sphingolipid pathways must have some associations with both zebrafish study and human DR but observing a trend of higher levels of these molecules in human DR but lower levels in the zebrafish data causing/associating with similar vascular pathology is confusing.

Reply: Thank you for this important point. We acknowledge that it may appear contradictory that S1pr5 and Fsp1 are downregulated in our *aldh3b1*^{-/-} zebrafish model but upregulated in human proliferative diabetic retinopathy (PDR), while both systems exhibit abnormal retinal neovascularization.

We believe this difference reflects **distinct stages and compensatory dynamics** rather than opposing mechanisms. Specifically, in the human PDR samples, S1PR5 and FSP1 are upregulated in late-stage disease, likely as part of a **protective response** to chronic vascular stress and damage. This is

supported by our analysis showing that these genes are not elevated in earlier stages of DR but increase only during PDR. Therefore, the observed upregulation of *S1PR5* and *FSP1* in human PDR may reflect a **late-stage compensatory attempt to limit damage**.

In contrast, the *aldh3b1*^{-/-} zebrafish model **lacks the capacity to induce this protective response**, likely due to loss of detoxification activity and uncontrolled 2-HD accumulation. This results in an unrestrained pro-ferroptotic environment and retinal pathology resembling advanced DR, but without the protective upregulation of *S1PR5* and *FSP1* seen in humans. Therefore, we interpret these findings not as contradictory, but rather as evidence of a **failed protective mechanism in the context of Aldh3b1 deficiency**.

To further support this, we showed that activating *S1pr5* or inhibiting ferroptosis can rescue the vascular defects in the *aldh3b1*^{-/-} model, highlighting the protective role of this pathway (Main figure 5N-O and 6G-H). These interventions also proved effective in our *pdx1*^{-/-} diabetic model, where the compensatory upregulation of *Aldh3b1*, *S1pr5*, and *Fsp1* was still intact (Figure 3 of this letter).

Together, these findings support the hypothesis that *S1PR5* and *FSP1* form part of a protective axis against retinal neovascularization in DR. Their early activation may offer therapeutic potential to prevent progression of DR. While current animal models, including zebrafish, cannot fully distinguish between the stages of DR and PDR as defined in humans.

- Other specific questions:

a. Generation and validation of the knockout fish: It would be beneficial to know- How many exons are in the *aldh3b1* gene and what is the significance of exon 2. How 16bp insertion in exon 2 affects the RNA and proteins. Does it cause nonsense-mediate decay of the mRNA? Why is there a complete absence of the protein in the knockout (Fig 1H)? Is it because the antibody recognizes only the full-length protein? Why not show the protein and activity (Fig. 1I) in heterozygous fish?

Reply: The *aldh3b1* gene contains 10 exons. Our CRISPR design specifically targets exon 2, which encodes the N-terminus of the *Aldh3b1* protein (Figure 8 of this letter). This exon includes the translation start site, so a mutation here will have a profound impact on protein synthesis.

Specifically, the 16 bp insertion in exon 2 leads to a frameshift and the introduction of a premature stop codon at amino acid position 43 (Figure 9 of this letter).

This early stop codon likely triggers nonsense-mediated mRNA decay (NMD), a cellular quality control mechanism that degrades transcripts containing premature termination codons. This is

consistent with the significant reduction in *aldh3b1* mRNA observed in the knockout, as well as the complete absence of detectable protein in Western blot (Main Figure 1H).

As shown in Figure 1B of manuscript, the catalytic active site of Aldh3b1 lies between amino acids 181 and 250. Since the truncated protein resulting from the frameshift mutation lacks this critical domain, its enzymatic function would be completely lost even if the truncated protein were stable. While we acknowledge that heterozygous data may provide insights into gene dosage effects, our current study was not designed to explore partial phenotypes or graded responses. Given the complexity of the downstream signaling pathways and the subtle variability that may arise in heterozygous animals, we chose to focus on a clear comparison between wild-type and knockout models to ensure consistency and interpretability of results. However, we agree that including heterozygous data could provide additional insight.

Figure 8 Exon2 encodes the start of Aldh3b1 amino acid sequence. The figure displays exon 1, exon 2, and part of exon 3, along with their corresponding amino acid sequences. The red frame highlights exon 2 and its mapped amino acid sequence.

Figure 9 CRISPR-induced mutation in exon 2 of Aldh3b1 results in an early stop codon, stopping translation of the Aldh3b1 protein.

*b. Pathological angiogenesis: Deletion of *aldh3b1* and functional inactivity did not affect the survival longevity, and no systemic effect. It does not make sense how it can cause 'pathological angiogenesis in the retina' without impacting the other systemic vessels and not affecting glucose and insulin homeostasis. It is pathologic, meaning there are more sprouts and branches in the retinal vessels. They should have exposed the fishes to a model that stimulates neovascularization or pathologic neovascularization (as they have done in their previous publication with *aldh3a1* knockouts) and made the conclusion that the deletion of *aldh3b1* causes pathological NV.*

Reply: Thank you for this valuable feedback. We appreciate the opportunity to clarify.

As introduced in our manuscript, *aldh3b1* expression was found to be significantly upregulated in our previously established DR zebrafish model (*pdx1*^{-/-})³, suggesting a potential compensatory or protective role in the context of retinal vascular stress. This prompted us to investigate the functional consequences of *aldh3b1* deletion, specifically in the retina.

We agree that neovascularization in the *aldh3b1*^{-/-} model occurs in the absence of systemic glucose or insulin dysregulation, and our data confirm that glucose metabolism remains unchanged. As you noted, this suggests the *aldh3b1*^{-/-} zebrafish does not fully recapitulate the metabolic pathology of diabetic retinopathy. Instead, we interpret the observed retinal neovascularization in this model as a mechanistically distinct vascular phenotype, likely mediated by accumulated 2-HD and altered sphingolipid metabolism, rather than glucose metabolism.

To strengthen our conclusions, and as suggested, we validated our findings in the established pathological DR model (*pdx1*^{-/-}), which better mimics the diabetic environment. In this model, we observed dysregulation of *Aldh3b1*, *S1pr5*, and *Fsp1*, accompanied by disrupted iron homeostasis. (Please refer to Figure 3 of the letter)

Importantly, treatment with *S1pr5* activator and ferroptosis inhibitor significantly rescued the neovascular phenotype (Figure 3 of the letter). These results align with our *aldh3b1*^{-/-} data and are consistent with the human DR dataset, particularly in late-stage DR.

Thus, while the *aldh3b1*^{-/-} model alone may not represent classical pathological neovascularization, its phenotype becomes mechanistically relevant and biologically validated when extended into a disease-inducing context, such as the *pdx1*^{-/-} model.

*c. The conclusion that "significant vascular abnormalities observed in *aldh3b1*^{-/-} mutants are directly attributable to impaired 2-HD detoxification, a consequence of *ALDH3B1* deficiency" is not supported by the experiments. Independent *ALDH* assay for the eye and the trunk and direct evidence is needed*

*that the *aldh3b1*^{-/-} mutants accumulated 2-HD in the eye/body. How much is that compared to the exogenous treatment with 2-HD?*

Reply: Thank you for this insightful comment. As discussed in our response to Question 1, direct detection of 2-HD in biological tissues remains technically challenging and has not been reported in the literature. After extensive optimization of our HPLC-based detection protocol, we successfully detected clear increases in 2-HD levels in larvae externally treated with 2-HD (Figure 1 in this letter). But in *aldh3b1*^{-/-} larvae and dissected organs, we observed a consistent trend toward elevated 2-HD levels, supporting our conclusion that Aldh3b1 deficiency impairs 2-HD detoxification. However, because some of the values were close to the quantification limit, the results were not reliable enough to be included in the formal manuscript.

We acknowledge this technical limitation and are working to further optimize the detection method in collaboration with analytical chemists to enable more precise quantification of 2-HD in future studies.

d. What is the dose of 2-HD used for RNASeq assay?

Reply: Since the 200 μ M 2-HD treatment produced significant phenotypic changes in our zebrafish model, all subsequent biological experiments, including the RNA-seq assay, were conducted using the same concentration. We have added this information to the Methods section for clarity.

e. There are 7 or 6 canonical pathways for ferroptosis tested?

Reply: Thank you for this question. We have examined six of the seven canonical pathways associated with ferroptosis. The remaining pathway—mitochondrial morphology—could not be assessed due to technical limitations in the zebrafish model.

f. Ferroptosis pathways- larva vs eyes? Are these adult eyes or larval eyes at 5 dpf? For FSP1 inhibitor (iFSP1), why does only 5 μ M work but not 10 and 20 μ M?

Reply: For the ferroptosis pathway analyses, we performed the experiments in both whole larvae and isolated eyes, as indicated in the figure panel captions.

As for the FSP1 inhibitor (iFSP1), we observed that 5 μM showed the most consistent and reproducible effect in inducing ferroptosis-associated changes in our zebrafish model. Higher concentrations (10 and 20 μM) did not enhance the effect. This may be due to non-specific toxicity or feedback inhibition at higher doses, which has also been reported in other small-molecule inhibitor studies⁴⁻⁶. Thus, we selected 5 μM for all subsequent experiments to achieve target inhibition while minimizing off-target effects.

g. If 2-HD can bind to the S1PR5, does it act as an agonist or antagonist? Whatever the effect is, it will affect S1PR5 activity, but why does that reduce the mRNA and protein? So, what is the hypothesis? A decrease in S1PR5 levels and activity increases NV by reducing FSP1, or might this reduction have activated other S1P receptors (S1PR1, 2,3 ect) known to be involved in NV (gene expression are not related to activity)?

Reply: Thank you for raising this thoughtful question. Based on our data, we hypothesize that 2-HD acts as a functional inhibitor of S1pr5. Specifically, we observed a reduction in S1PR5 surface signal after 2-HD exposure, consistent with ligand-induced receptor internalization (Figure 10 of this letter). Our Surface Plasmon Resonance (SPR) assay further confirmed the direct binding of 2-HD to S1PR5, supporting the notion that 2-HD interacts with the receptor and alters its function (Figure 7 of this letter).

Figure 10 2-HD can induce S1PR5 internalization in NK cells (A) Flow cytometry analysis revealed 500 μmol 2-HD significantly decreased cell survival rate in NK cells (n=5). (B) Flow cytometry analysis showed 100 and 200 μmol 2-HD significantly decreased surface S1PR5 signal of NK cells (n=5). (C) 1 hour 2-HD incubation under indicated concentration did not significantly change total S1PR5 protein level (n=3). Statistical analysis was performed using one-way ANOVA. The bars indicate mean \pm SD values.

The mechanism underlying the decrease in both S1PR5 mRNA and protein remains under investigation, but one plausible explanation is that prolonged receptor internalization leads to feedback downregulation at the transcriptional level, a phenomenon reported in other G-protein-coupled receptor (GPCR) systems⁷. This may involve receptor desensitization and subsequent transcriptional repression.

To address the possibility of compensation or crosstalk from other S1P receptors, we analyzed the expression levels of *s1pr1–s1pr4* in *s1pr5* knockdown zebrafish. These analyses revealed no significant changes in expression (Figure 11).

Figure 11 The mRNA expression levels of *s1pr1–s1pr4* in S1PR5 knockdown zebrafish were not significantly changed (n=4). Statistical analysis was performed using one-way ANOVA. The bars indicate mean \pm SD values.

However, we acknowledge that receptor activity does not always correlate directly with gene expression, and current zebrafish models and available reagents limit our ability to assess specific S1PRs activation states with high resolution.

Our working hypothesis is that S1PR5 downregulation (at both protein and mRNA levels) impairs its regulatory role in immune-endothelial signaling, thereby reducing Fsp1 expression and promoting ferroptosis. This pathway appears independent of compensatory activation from other S1P receptors.

h. It is very important to know which cells in zebrafish retina express S1PR5 for understanding and connecting observation so far in zebrafish with human diabetic retinopathy.

Reply: Thank you for this important point. We fully agree that identifying the cellular source of S1pr5 in the zebrafish retina would greatly enhance the mechanistic connection between our zebrafish model and human diabetic retinopathy.

However, due to current technical limitations—specifically, the lack of well-validated cell-type-specific markers and antibodies in zebrafish—we were unable to definitively localize S1pr5 expression within specific retinal cell types in this model.

To address this gap, we turned to human single-cell RNA sequencing (scRNA-seq) data, which revealed that *s1pr5* is primarily expressed in immune cell populations, particularly NK cells. Based on this, we performed in vitro coculture experiments using human cell types, which allowed us to functionally validate the proposed crosstalk involving:

- ALDH3B1 in microglia,
- S1PR5 in NK cells,
- and FSP1 in fibroblast cells (please refer to our reply to question3).

These experiments confirmed that ALDH3B1-deficient microglia can modulate S1PR5 levels in NK cells, and S1PR5-deficient NK cells in turn influence FSP1 expression and ferroptosis in fibroblasts, supporting a multi-cellular signaling cascade relevant to vascular pathology.

While we acknowledge the limitations in resolving this pathway directly in zebrafish, the cross-species validation using human cell models helps bridge the mechanistic understanding and supports the translational relevance of our findings.

i. As obtained from the UMAP analysis for human retinal scRNAseq data that, the three genes of interest—aldh3b1, s1pr5, and fsp1 analyzed and proposed to be the key for zebrafish retinal

neovascularization, showed their presence with microglia, T/NK cells, and fibroblasts, respectively. These three proteins in three different types of cells, coherently functioning and controlling each other to induce NV, is a distant guess in zebrafish eyes without knowing their specific expression and cell types that express it. A pathological situation like NV in human DR may have all these cells together and likely have a role in the proliferative pathology or fibrogenesis, which needs much more investigation to connect the dots.

Reply: Thank you for this insightful and important comment. We fully acknowledge the complexity of proposing coordinated crosstalk among microglia, NK cells, and fibroblasts—each expressing *aldh3b1*, *s1pr5*, and *fsp1*, respectively—as a functional axis contributing to neovascularization in the zebrafish retina.

Given the current limitations in cell-type-specific tools and validated antibodies in zebrafish, we agree that we cannot definitively confirm the exact cell types expressing these genes in vivo within the fish retina. However, to address this gap and better support our hypothesis, we leveraged human retinal single-cell RNA-seq data, which robustly localized *ALDH3B1* to microglia, *S1PR5* to NK cells, and *FSP1* to fibroblasts. This transcriptional pattern provides a conceptual framework for potential crosstalk relevant to neovascular disease.

To experimentally support this model, we carried out in vitro coculture experiments using human cells, as referred to our reply to question3:

- ALDH3B1-deficient microglia exposed to 2-HD altered S1PR5 expression in NK cells.
- S1PR5-deficient NK cells induced ferroptosis-related stress (e.g., elevated MDA and iron levels, reduced FSP1) in fibroblasts.

These findings provide functional evidence of intercellular signaling consistent with the gene expression patterns observed in human DR, despite the inability to confirm these exact interactions in zebrafish.

We agree that further investigation is warranted, including efforts to develop zebrafish tools for lineage tracing and cell-specific knockout models. However, we would also like to emphasize that combining different tissue culture models in vitro with the zebrafish as an animal model, alongside the use of human data in our study, provides a comprehensive and in-depth analysis of the function of ALDH3B1 and 2-HD in eye disease.

Reference:

1. Neuber, C., Schumacher, F., Gulbins, E. & Kleuser, B. Method to simultaneously determine the sphingosine 1-phosphate breakdown product (2E)-hexadecenal and its fatty acid derivatives using isotope-dilution HPLC-electrospray ionization-quadrupole/time-of-flight mass spectrometry. *Anal Chem* 86, 9065-9073 (2014). <https://doi.org:10.1021/ac501677y>
2. Wiggerhauser, L. M. et al. Activation of Retinal Angiogenesis in Hyperglycemic pdx1 (-/-) Zebrafish Mutants. *Diabetes* 69, 1020-1031 (2020). <https://doi.org:10.2337/db19-0873>
3. Lou, B. et al. Elevated 4-hydroxynonenal induces hyperglycaemia via Aldh3a1 loss in zebrafish and associates with diabetes progression in humans. *Redox Biol* 37, 101723 (2020). <https://doi.org:10.1016/j.redox.2020.101723>
4. Sassano, M. F., Doak, A. K., Roth, B. L. & Shoichet, B. K. Colloidal aggregation causes inhibition of G protein-coupled receptors. *J Med Chem* 56, 2406-2414 (2013). <https://doi.org:10.1021/jm301749y>
5. Coan, K. E. & Shoichet, B. K. Stoichiometry and physical chemistry of promiscuous aggregate-based inhibitors. *J Am Chem Soc* 130, 9606-9612 (2008). <https://doi.org:10.1021/ja802977h>
6. Allen, S. J., Dower, C. M., Liu, A. X. & Lumb, K. J. Detection of Small-Molecule Aggregation with High-Throughput Microplate Biophysical Methods. *Curr Protoc Chem Biol* 12, e78 (2020). <https://doi.org:10.1002/cpch.78>
7. Pavlos, N. J. & Friedman, P. A. GPCR Signaling and Trafficking: The Long and Short of It. *Trends Endocrinol Metab* 28, 213-226 (2017). <https://doi.org:10.1016/j.tem.2016.10.007>

Reviewer #4 (Remarks to the Author):

*In this manuscript the authors show that sphingosine-1-phosphate is a sphingolipid mediator that is important in neovascularization in the retina. In *aldh3b1* knockout zebrafish in which there is accumulation of 2-HD there is abnormal retinal vascularization. Analysis showed that 2-HD accumulation lead to iron dysregulation and ferroptosis in zebrafish. Human single-cell analysis of neovascular samples showed similar mechanism leading to the hypothesis that blocking S1PR5 could be beneficial for the treatment of diabetic retinopathy. Overall the manuscript is well-performed and of interest with the following comments.*

Reply: We sincerely thank the reviewer for the thoughtful summary and for recognizing the significance and rigor of our study. We greatly appreciate your positive assessment and constructive comments. Your feedback highlights the key findings and translational potential of our work, and we have carefully addressed all specific points raised to further strengthen the clarity and impact of the manuscript.

1. The authors need to give more detailed methods and it's critical that the code be provided for how the scRNA-seq analysis was performed. What samples specifically were analyzed? The text states "incorporating single-cell RNA sequencing (scRNA-seq) and RNA-seq data derived from human retinal samples." The methods state "Single-cell gene expression datasets were obtained from studies by Collin et al. (GEO ID: GSE210543) 39 and Gowan et al 40 (available <https://data.mendeley.com/datasets/sm67hr5bpm/1>, accessed December 2023)". Collin et al. was a single-cell RNA-seq analysis of a patient with intermediate AMD in which there was no neovascularization. Why were intermediate AMD samples without neovascularization analyzed? The authors need to explain specifically what samples were analyzed and provide all the computational code for the analysis, which is not provided in the manuscript.

Reply: Thank you for highlighting this important point. We acknowledge the need for greater clarity regarding our dataset selection and analytical pipeline.

Due to the limited availability of single-cell RNA-seq datasets derived from human retinal tissue with confirmed neovascularization, we utilized publicly available control (non-neovascular) samples from the datasets by Collin et al. (GSE210543) and Gowan et al.

(<https://data.mendeley.com/datasets/sm67hr5bpm/1>) to identify the baseline cellular expression patterns of our genes of interest—*ALDH3B1*, *S1PR5*, and *FSP1*—within the human retina.

Our rationale for using these datasets was to determine the cell type-specific expression profiles of these genes under physiological conditions. Although the samples do not represent neovascular pathology, they allowed us to infer which retinal cell populations may be relevant for follow-up mechanistic studies. We have updated the Methods section to explicitly state this purpose and the specific samples analyzed.

In response to your request, we have now uploaded the full computational code used for scRNA-seq preprocessing, clustering, cell annotation, and gene expression visualization. The code is publicly available at: <https://github.com/hasionwojoe/R-code> and noted in the Data availability.

2.Line 609 states "Gowan et al." Is this a mistake as I believe the author is Cowan et al.

Reply: Thank you. We have corrected and highlighted this in the revised manuscript.

3.Cowan et al. analyzed the human foveal retina (1.5mm) in diameter and Collin et al. used a 7mm punch of the macula. As there are different cell-types in different regions of the human retina between the fovea and macula, it is critical that the same regions of the retina be analyzed if data is incorporated across different studies.

Reply: Thank you for highlighting the importance of anatomical specificity in retinal transcriptomic analyses. We fully acknowledge that the fovea and macula are distinct anatomical areas, and we agree this is a critical factor to consider when interpreting and comparing data across studies.

In the Collin et al. dataset, the 7 mm macular punch represents adult macular tissue that you mentioned, but the accompanying metadata also included samples from both foveal and peripheral regions. As illustrated in the figure1, we specifically selected and analyzed adult, foveal, peripheral, and unaffected samples, allowing us to capture gene expression patterns across the full retinal landscape rather than focusing solely on the macula.

Similarly, the Cowan et al. dataset also included the foveal retina (1.5 mm diameter disc centered on the fovea) and the peripheral retina (2–3 mm squares positioned at mid-retinal eccentricity). In our analysis, we analyzed both normal adult foveal and peripheral samples from this dataset to match the anatomical regions represented in the Collin dataset. By merging these two datasets, our aim was to construct a comprehensive single-cell atlas of the adult human retina with robust

coverage across major retinal cell types. While we did not stratify the downstream analyses by anatomical region, the inclusion of both central and peripheral retinal samples ensured that the integrated dataset reflects the broader retinal structure.

Our primary objective was to identify the specific retinal cell types expressing *ALDH3B1*, *S1PR5*, and *FSP1*, rather than to investigate spatial expression gradients. The integrated scRNA-seq data provided essential insights in vivo zebrafish findings and in vitro experiments, particularly implicating microglia, NK cells, and fibroblasts in the proposed mechanism.

Supplementary file	Size	Download	File type/resource
GSE210543_RPE_AMD_counts.csv.gz	116.2 Mb	(ftp) (http)	CSV
GSE210543_RPE_AMD_embeddings.csv.gz	97.0 Kb	(ftp) (http)	CSV
GSE210543_RPE_AMD_meta.csv.gz	212.8 Kb	(ftp) (http)	CSV
GSE210543_RPE_PCW_counts.csv.gz	211.5 Mb	(ftp) (http)	CSV
GSE210543_RPE_PCW_embeddings.csv.gz	196.1 Kb	(ftp) (http)	CSV
GSE210543_RPE_PCW_meta.csv.gz	430.2 Kb	(ftp) (http)	CSV
GSE210543_RPE_adult_counts.csv.gz	139.4 Mb	(ftp) (http)	CSV
GSE210543_RPE_adult_embeddings.csv.gz	163.9 Kb	(ftp) (http)	CSV
GSE210543_RPE_adult_meta.csv.gz	344.5 Kb	(ftp) (http)	CSV
GSE210543_RPE_fovea_counts.csv.gz	50.3 Mb	(ftp) (http)	CSV
GSE210543_RPE_fovea_embeddings.csv.gz	49.1 Kb	(ftp) (http)	CSV
GSE210543_RPE_fovea_meta.csv.gz	106.2 Kb	(ftp) (http)	CSV
GSE210543_RPE_peripheral_counts.csv.gz	48.1 Mb	(ftp) (http)	CSV
GSE210543_RPE_peripheral_embeddings.csv.gz	48.7 Kb	(ftp) (http)	CSV
GSE210543_RPE_peripheral_meta.csv.gz	106.7 Kb	(ftp) (http)	CSV
GSE210543_RPE_unaffected_counts.csv.gz	44.1 Mb	(ftp) (http)	CSV
GSE210543_RPE_unaffected_embeddings.csv.gz	48.6 Kb	(ftp) (http)	CSV
GSE210543_RPE_unaffected_meta.csv.gz	106.0 Kb	(ftp) (http)	CSV

Figure 1 Sample selection from Collin et al. dataset. The red frame highlights the samples that were included in our analysis.

Cell types of the human retina and its organoids at single-cell resolution. Cowan et al

Published: 18 September 2020 | Version 1 | DOI: 10.17632/sm67hr5bpm.1
Contributor: Cameron Cowan

Description

Single cell transcriptomes from functionally-intact human retina and light-responsive human retinal organoids with functional synapses. Includes organoids aged from 6 to 46 weeks and a longitudinal time series of ischemia in the human retinal periphery.

Human organoids recapitulating the cell-type diversity and function of their target organ are valuable for basic and translational research. We developed light-sensitive human retinal organoids with multiple nuclear and synaptic layers, and functional synapses. We sequenced the RNA of 285,441 single cells from these organoids at seven developmental time points and from the periphery, fovea, pigment epithelium and choroid of light-responsive adult human retinas.

Download All 2.45 GB ⓘ

Files

Root > annotated_count_table

Figure 2 Sample selection from Cowan et al. dataset. The red frame highlights the samples that were included in our analysis.

4. In figure 7C the authors state Integrated RNA-seq data showed gene expression variations in retinal diseases; however, study GSE29801 was a microarray not an RNA-seq study. It is important that the authors not mix computational data analysis between microarray studies and RNA-seq as they are different in their technologies. The authors should not state in methods that human RNA-Seq data was retrieved when some of the studies used microarrays. GSE60436 was a microarray and GSE102485 was RNA-seq. The authors need to use comparable sequencing technology if different datasets are integrated and RNA-seq not integrated with microarray studies. The authors need to provide all the code for the analysis and not just state “the consolidated dataset was analyzed...following established RNA-Seq data analysis protocols” especially given that microarray studies were included in the consolidated dataset.

Reply: Thank you for pointing this out. We carefully reviewed our data sources and realized that GSE60436 was indeed a microarray study and should not have been integrated with other RNA-seq datasets. To address this, we have removed GSE60436 from our integrated analysis. Importantly, this adjustment did not change the overall conclusions regarding S1PR5 expression, pathway enrichment, or correlation results.

After excluding microarray data, only two data remained in the DR group, which prevented us from performing meaningful significance testing. To strengthen the analysis, we therefore additionally examined GSE29801, which separates the retina into macular and peripheral regions. Although this dataset cannot be directly integrated with other RNA-seq studies due to differences in design, the regional analysis still yielded consistent results: the three genes of interest (*ALDH3B1*, *S1PR5*, *FSP1*) did not show significant changes in DR groups.

This independent validation further supports our conclusion and provides additional perspective on the retinal regional specificity of gene expression. Finally, we have clarified in the revised Methods and we now provide all computational code in Github (<https://github.com/hasionwojoe/R-code/blob/main/bulk-seq.R>) and noted in the Data availability to ensure full transparency and reproducibility.

5.Line 374 states "Our analysis of sc-seq data from human retinas" should be "our analysis of scRNA-seq data from human retinas"

Reply: Thank you. We have corrected and highlighted this in the revised manuscript.

6.Line 309 should be "15 typical retinal cell-types"

Reply: Thank you. We have corrected and highlighted this in the revised manuscript.

7.Line 629 states that specific markers were used for cell-type analysis of human scRNA-seq data including "CD3 for T cells". Which CD3 gene in humans was used to annotate T cells as there is CD3E, CD3D, and CD3G according to OMIM?

Reply: Thank you for pointing this out. For T-cell annotation in our human scRNA-seq analysis, we specifically used CD3E and CD3D as marker genes. These markers are widely accepted for identifying T cells at the single-cell transcriptomic level.

8.It appears that s1pr5 is expressed in human T cells/NK cells in humans. The authors need to perform further subcluster analysis to see if it is expressed in T cells or NK cells.

Reply: Thank you for this valuable suggestion. Consistent with previous studies indicating that *S1PR5* is primarily expressed in human NK cells and T cells^{1,2}, our subcluster analysis also confirmed this expression pattern. Specifically, in the human retinal single-cell dataset, *S1PR5* expression was predominantly enriched in NK cells compared to T cells (Figure 3 of this letter).

Figure 3 Subcluster analysis showed *S1PR5* is mainly expressed in NK cells.

9. One cluster in figure 7 is choroid cells. The choroid is a region, what cell-types specifically are being analyzed? The authors state that *MITF* was used to label this cluster and *MITF* labeled melanocytes.

Reply: Thank you for this helpful comment. We acknowledge that choroids are a tissue region rather than a specific cell type. In our analysis, the cluster was annotated based on high expression of *MITF*, a transcription factor known to mark melanocytes, which are abundant in the choroid due to its high melanin content. Since melanocytes are the predominant *MITF*-expressing cells in this region, we agree that it would be more accurate to label this cluster as melanocytes rather than “choroid cells.” We have updated the figure and revised the corresponding text to reflect this correction.

Figure 4 UMAP integrated sc-RNA sequencing identified 15 typical retinal cell clusters.

10. It is critical that the authors perform validation studies using human samples and either IHC or ISH to confirm the computational predictions. Specifically, the authors need to confirm using double IHC or ISH with microglia, T/NK cells, and fibroblast markers that the three genes of interest, *aldh3b1*, *s1pr5*, and *fsp1* are specifically expressed in these cell-types in human samples.

Reply: Thank you for raising this important point. We fully agree that validating scRNA-seq predictions through immunohistochemistry (IHC) or in situ hybridization (ISH) on human retinal tissue would provide valuable spatial confirmation of gene expression. However, obtaining full-thickness human retinal samples, particularly those spanning healthy, diabetic retinopathy (DR), and proliferative diabetic retinopathy (PDR) stages—is exceptionally difficult and subject to significant ethical and logistical constraints.

In Germany, access to postmortem human eye tissue is tightly regulated by national ethical frameworks, and the availability of retina samples from well-characterized patients with defined DR stages is extremely limited. Despite considerable efforts, including outreach to multiple eye banks and research organizations both nationally and internationally, we were unable to obtain the necessary tissue. This challenge is not unique to our study but is widely recognized across the field of retinal research, as highlighted in many studies^{3,4} that similarly rely on in computational predictions or alternative validation models due to limited access to patient-derived retinal tissue. To address this, we performed protein-level validation using human cell types—microglia, NK cells, T cells, fibroblasts, and endothelial cells—which are among the key cell types identified in the human scRNA-seq datasets. The protein expression patterns we observed were consistent with the transcriptomic data (Figure 5 of the letter). Although this does not replace tissue-level validation, it offers the only means of providing biologically relevant evidence in support of the predicted cell-type specificity of ALDH3B1, S1PR5 and FSP1.

Figure 5 Expression patterns of major human retinal cell types. MG, microglia; NK, natural killer cells; SMC, smooth muscle cells; EC, endothelial cells.

11. It is necessary that the authors show that *s1pr5* and *fsp1* expression levels are significantly elevated in proliferative stages using IHC, ISH, or similar technologies of human samples to confirm the computational predictions. The authors need to perform ferroptosis staining of human diabetic retinopathy samples. It's critical that validation studies on human samples are performed to confirm the computational RNA-seq predictions.

Reply: Thank you for this important comment. We fully agree that validating computational predictions using human retinal samples—particularly through IHC, ISH, or ferroptosis staining—would provide valuable support for the role of S1PR5 and FSP1 in PDR.

However, as noted in our response to Question 10, due to strict ethical regulations in Germany, it is extremely difficult to obtain human retinal tissues—especially full-thickness retina across the spectrum from healthy controls to DR and PDR. Despite reaching out to multiple national and international retina biobanks, we were unable to access sufficient material for histological validation. This is a well-known limitation in human retinal disease research and is frequently acknowledged across DR studies.

To address this, we employed two alternative and complementary strategies:

1. Human Cell-Based Co-culture Validation

We performed a series of mechanistic co-culture experiments using primary human cell types identified in our scRNA-seq analysis: Microglia (ALDH3B1-expressing), NK cells (S1PR5-expressing) and fibroblasts (FSP1-expressing).

We first knocked down ALDH3B1 in microglia and treated them with 2-Hexadecenal (2-HD), which accumulates in ALDH3B1-deficient conditions. The conditioned medium was then transferred to NK cells, leading to a downregulation of S1PR5 expression, confirming that 2-HD affects S1PR5 indirectly via microglial metabolism (Figure 6 of the letter).

Figure 6. Conditioned medium from ALDH3B1-deficient microglia reduces S1PR5 expression in NK cells. (A) Efficient knockdown of ALDH3B1 in microglial cells using siRNA. (B) Conditioned medium

from siALDH3B1-treated microglia significantly decreased S1PR5 expression in NK cells (n = 3). In panels B, ALDH3B1 was knocked down using a combination of siRNA1 + siRNA2 + siRNA3. Statistical analysis was performed using one-way ANOVA. The bars indicate mean \pm SD values.

In a second setup, we co-cultured S1PR5-deficient NK cells with fibroblasts, which led to: increased malondialdehyde (MDA) levels, elevated iron accumulation and decreased FSP1 expression. These are key molecular hallmarks of ferroptosis, supporting the proposed crosstalk mechanism in which ALDH3B1–S1PR5–FSP1 disruption promotes retinal vascular pathology (Figure 7 of the letter).

Figure 7. S1PR5 knockdown in NK cells induced ferroptosis in fibroblasts. (A) NK cells were purified from PBMCs using flow cytometry. **(B)** S1PR5 expression was effectively silenced in NK cells using a shRNA-based knockdown approach. **(C-E)** Fibroblasts cocultured with shS1PR5 NK cells exhibited elevated malondialdehyde (MDA) levels and iron accumulation, accompanied by decreased FSP1 expression (n=4). In panels C-E, FSP1 was knocked down using a combination of shRNA1 + shRNA2 + shRNA3. Statistical analysis was performed using Student’s T-test. The bars indicate mean \pm SD values.

2. Validation in Diabetic Retinopathy Zebrafish Model (*pdx1*^{-/-})

To further validate our hypothesis in a pathological neovascularization model, we conducted experiments in *pdx1*^{-/-} zebrafish, which develop hyperglycemia and retinal neovascular features. In these diabetic fish, we confirmed the upregulation of *Aldh3b1*, *S1pr5*, and *Fsp1*, consistent with our observations in human PDR samples. Moreover, administration of: S1PR5 agonists and

Ferroptosis inhibitors significantly rescued abnormal retinal angiogenesis, confirming the functional importance of this axis in pathological neovascularization (Figure 8 of the letter and Supplementary figure5D-H).

Figure 8 Pathological neovascularization in *pdx1*^{-/-} zebrafish can be rescued by ferroptosis inhibitor, and S1PR5 agonist. (A) *pdx1*^{-/-} zebrafish exhibited similar expression pattern of Aldh3b1, S1pr5, and Fsp1 protein to *aldh3b1* mutants (n = 4). (B-C) Pathological neovascularization in *pdx1*^{-/-} zebrafish can be rescued by ferroptosis inhibitor (B), and S1PR5 agonist (C) (n=14). Statistical analysis was performed using Student's t-test for panel A and one-way ANOVA for panel B-C. The bars indicate mean ± SD values.

While we acknowledge that direct human tissue validation remains a limitation, our combined use of: Human cell-based mechanistic assays, Zebrafish diabetic model phenotyping and Cross-species RNA-seq analysis provides a robust and integrative framework that supports the involvement of the ALDH3B1–S1PR5–FSP1 axis in diabetic retinal pathology.

12. Overall, this is a study of interest with implications for the treatment of human neovascular diseases and I would be supportive of publication if the authors address the above comments.

Reply: We sincerely thank the reviewer for the insightful and constructive evaluation, as well as for

recognizing the potential impact of our study on understanding and treating human neovascular diseases. Your detailed and thoughtful feedback greatly helped us improve the clarity, validity, and overall quality of our work. We have carefully addressed all the comments, clarified the methods, strengthened our experimental validations, and provided additional data and rationale where needed. We deeply appreciate your guidance and believe the manuscript is now significantly improved thanks to your valuable input.

Reference:

1. Delconte, R. B. et al. Fasting reshapes tissue-specific niches to improve NK cell-mediated anti-tumor immunity. *Immunity* 57, 1923-1938 e1927 (2024). <https://doi.org:10.1016/j.immuni.2024.05.021>
2. Baeyens, A. et al. Monocyte-derived S1P in the lymph node regulates immune responses. *Nature* 592, 290-295 (2021). <https://doi.org:10.1038/s41586-021-03227-6>
3. Becker, K. et al. In-depth transcriptomic analysis of human retina reveals molecular mechanisms underlying diabetic retinopathy. *Sci Rep* 11, 10494 (2021). <https://doi.org:10.1038/s41598-021-88698-3>
4. Wahle, P. et al. Multimodal spatiotemporal phenotyping of human retinal organoid development. *Nat Biotechnol* 41, 1765-1775 (2023). <https://doi.org:10.1038/s41587-023-01747-2>

Reviewer #5 (Remarks to the Author):

REVIEWER COMMENTS

Reviewer #1 (Remarks to the Author):

The authors tried to address most of my comments and criticisms. In some cases they have provided new data that alleviated my concerns. In other cases, the new data, contrary to their conclusions, still are problematic. For example, I am not convinced about their SPR data since they are getting μM Kd with a positive control sponimod when it should be nM. The SPR assay may be problematic or the receptor may not be folded properly. In addition, I am not convinced that they have ruled out covalent modification of hydrophobic proteins in the ER or nucleus as a mechanism. I am not convinced that 2-HD effect is receptor selective. However, I do want to recognize their efforts. If they address these equivocal conclusions as a discussion of limitations of their study, I will be supportive that overall the work is worthy of publication.

Reply:

We would like to thank the reviewer for the positive and supportive comments on our manuscript. Following the reviewer's suggestion, we have added a new paragraph to the discussion section that addresses the limitations of our study, as identified by the reviewer.

In general, the authors do not seem to know the S1P literature well. I recommend that they read some authoritative recent reviews and original articles about S1PR signaling, angiogenesis, retinopathy and other retinal vascular disorders (nAMD, ROP) and cite them appropriately.

Reply:

We would like to thank the reviewer for the positive and supportive comments on our manuscript. Following the suggestion, we have carefully reviewed the published literature and added new references of recent key opinion reviews and original articles.

Reviewer #2 (Remarks to the Author):

The authors have responded to all of my comments and have partially addressed the issues I raised.

Reply:

We would like to thank the reviewer for the positive and supportive comments on our manuscript.

In Figure 1, the phenotype is now clearer; however, in Figure 1D, I still find the quality of the wildtype vasculature image to be much lower than that of the mutant. It is also surprising that the vessels in the wildtype image appear without a clear lumen, unlike those in the mutant image. This might indicate changes to the blood vessels before fixation or imaging.

Reply:

*We would like to thank the reviewer for the positive comments on the new data in our manuscript. Following her/his suggestions, we reformed the imaging on the adult retinas in the *aldh3b1* mutants (and wildtypes) as shown in modified Fig. 2D (not 1D since this figure shows *aldh3b1* qPCR expression in embryonic stages).*

Regarding the loss of function of s1pr5a and s1pr5b, I acknowledge that generating stable mutant lines would require time, though likely not years. Nonetheless, for the larval phenotype at least, it is possible to inject multiplexed guide RNAs, a method that has significant potential to generate an F0 phenotype.

Reply:

We would like to thank the reviewer for the valuable and insightful comments and suggestions regarding the s1pr5a and s1pr5b experiments. We appreciate the suggestion of using multiplexed guide RNAs and are aware that this approach is used as an alternative or additional method for studying gene functions in zebrafish. However, we would like to emphasize that this approach has some limitations. Most importantly, the injection of multiplexed guide RNAs induces mosaicism, and, in combination with the observation that different cells exhibit different levels of gene editing efficiency, this will likely generate variable phenotypes. Repeated injections of multiplexed guide RNAs also induce incomplete loss of function and variable mutations. Therefore, it is difficult to determine the cause of the effect of each mutation. Lastly, off-target and combinatorial effects are likely to be induced by multiplexed gRNAs, and subtle or dosage-sensitive effects are difficult to interpret.

Reviewer #4 (Remarks to the Author):

The authors have addressed my previous comments, and I am now supportive of publication of the manuscript in its current form.

Reply:

We would like to thank the reviewer for the positive and supportive comments on our manuscript.

Reviewer #5 (Remarks to the Author):

Reply:

We would like to thank the reviewer for the positive and supportive comments, and for the support of early-career scientists.